# MC-LLaVA: Multi Concept Personalized Vision Language Model

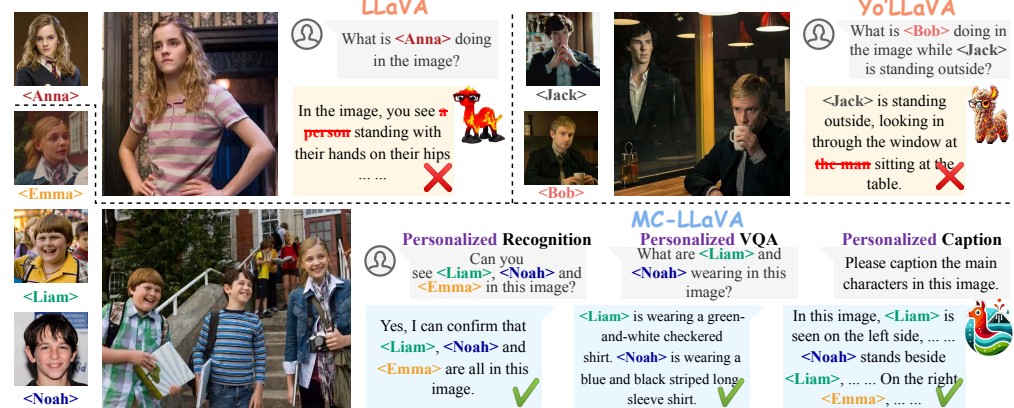

Figure 1: LLaVA fails to understand user-provided concepts. Existing methods like Yo'LLaVA mainly focus on single-concept personalization and can not generate accurate, personalized responses about multi-concepts. The proposed MC-LLaVA learns multiple concepts and can perform accurately in multi-concept scenario across various tasks such as recognition, VQA, and captioning.

## ABSTRACT

Current vision-language models (VLMs) show exceptional abilities across diverse tasks, such as visual question answering. To enhance user experience, recent studies investigate VLM personalization to understand user-provided concepts. However, they mainly focus on single-concept personalization, neglecting the existence and interplay of multiple concepts, which limits real-world applicability. This paper proposes the first multi-concept personalization paradigm, MC-LLaVA. Specifically, MC-LLaVA employs a multi-concept instruction tuning strategy, effectively integrating multiple concepts in a single training step. To reduce the costs related to joint training, we propose a personalized textual prompt that uses visual token information to initialize concept tokens. Additionally, we introduce a personalized visual prompt during inference, aggregating location confidence maps for enhanced recognition and grounding capabilities. To advance multi-concept personalization research, we further contribute a high-quality instruction tuning dataset. We carefully collect images with multiple characters and objects from movies and manually generate question-answer samples for multi-concept scenarios, featuring superior diversity. Comprehensive qualitative and quantitative experiments demonstrate that MC-LLaVA can achieve impressive multi-concept personalized responses, paving the way for VLMs to become better user-specific assistants. The code and dataset will be released.

## 1 INTRODUCTION

Over the past few years, large language models (LLMs) Achiam et al. (2023); Bai et al. (2023); Yang et al. (2023a); Touvron et al. (2023) have made significant advancements, proving their effectiveness in various applications and transforming the way humans interact with machines. In line with this trend, many vision-language models (VLMs) Liu et al. (2024); Li et al. (2023a); Wang et al. (2024); Bai et al. (2024) have been proposed to connect vision encoders with LLMs for various vision-

language tasks Deng et al. (2024); Zhou et al. (2024); Li et al. (2024); Lin et al. (2024) such as visual question answering. Despite their success, VLMs face challenges when personalized responses are required, such as answering visual questions based on user-provided concepts. For example, given images with a concept ⟨Anna⟩, VLMs fail to generate sentences with its identifier, as illustrated in Fig. 1. This limitation hinders the smooth integration of VLMs into our daily lives.

Although some methods Nguyen et al. (2024); Alaluf et al. (2025); Hao et al. (2024a) have produced impressive results in VLM personalization, they mainly concentrate on single-concept personalization. However, in real-world scenarios, vision-language tasks often involve multiple concepts, which is crucial for the effective deployment of personalized VLMs. From this perspective, personalizing multiple concepts while ensuring that VLMs respond accurately can be challenging for current methods. For example, Yo'LLaVA Nguyen et al. (2024) faces two main challenges when directly applied to multi-concept scenarios. First, because it trains each concept separately, merging

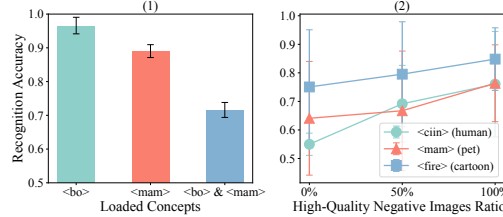

Figure 2: **Case studies utilizing various concepts from the Yo'LLaVA dataset.** The left panel shows the limitations of separately trained Yo'LLaVA models, while the right panel emphasizes the significance of high-quality negative samples for the training process of Yo'LLaVA.

parameters for different concepts leads to severe performance degradation due to concepts confusion (see Fig. 2(1)). Second, Yo'LLaVA learns concept tokens and classifier heads from scratch, resulting in a strong reliance on high-quality negative samples (see Fig. 2(2)). As the number of concepts increases, the demand for negative samples also rises, making data collection more challenging and requiring the model to spend additional time fitting the data. Consequently, multi-concept personalization of VLMs incurs significantly higher manual labor and training costs.

To solve the abovementioned problems, we introduce a novel method called MC-LLaVA, which ensures the accurate generation of personalized responses based on multiple concepts. MC-LLaVA considers multiple concepts together in a single training step rather than treating them independently. To reduce the cost of joint training, we pass all concept images through the VLM vision encoder and projection layer, using projected vision embeddings to initialize the concept tokens in personalized textual prompts. Our experiments show that this initialization can accelerate training and reduce dependence on high-quality negative samples. Additionally, MC-LLaVA enhances the model's perception capabilities by introducing a personalized visual prompt. We aggregate the location confidence maps based on concept tokens to create the personalized visual prompt for VLMs.

To advance research in multi-concept personalization, datasets for training and testing are essential. Recent studies Alaluf et al. (2025); Nguyen et al. (2024) have developed datasets for personalized VLMs; however, these datasets focus only on evaluating single concepts. Furthermore, the types of questions and answers they address are limited to basic recognition and multiple-choice formats. The lack of datasets hinders the progress of multi-concept personalized VLMs. Therefore, we contribute a high-quality dataset by meticulously gathering images from concept-rich movies. We then utilize GPT-4o Achiam et al. (2023) to generate the initial question-answer samples and then manually refine the generated samples. Our dataset features diverse movie types and question-answer types. In total, our dataset includes approximately 2,000 images and 16,700 question-answer samples. To comprehensively evaluate multi-concept personalized VLMs, we assess MC-LLaVA across various tasks, including multi-concept recognition, visual grounding, question answering (QA), and captioning. Our dataset will facilitate future research in VLM personalization.

We summarize our contributions as follows:

- We introduce MC-LLaVA, the first method designed for multi-concept VLM personalization, which employ personalized textual and visual prompts to learn various concepts and generate tailored responses effectively, reducing reliance on high-quality negative samples.

- Queried GPT-4o approximately 100K times, we contribute a large-scale and high-quality dataset for training and testing multi-concept personalized VLMs.

- We perform thorough experiments on our dataset and two additional benchmarks, achieving state-of-the-art results among various tasks in single- and multi-concept personalization.

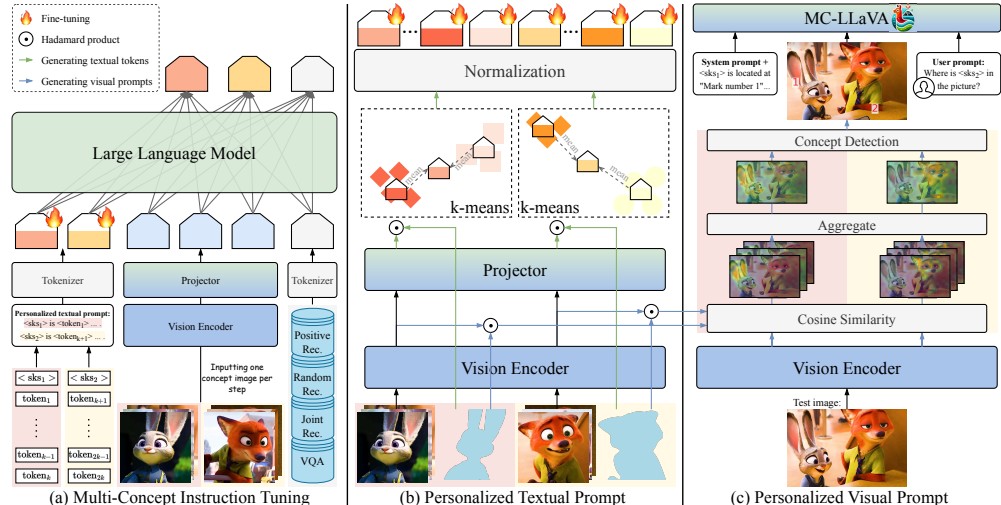

Figure 3: **Overview of MC-LLaVA.** (a) We use a multi-concept joint training strategy to learn the personalized textual prompts and classifier weights. (b) Given $m$ concepts, we utilize visual tokens obtained from K-means centroids to initialize the $m \times (k+1)$ concept tokens in personalized textual prompts, reducing costs associated with training. (c) During inference, we introduce personalized visual prompts by aggregating location confidence maps based on learned concept tokens.

## 2 RELATED WORK

**Personalized VLMs.** The emergence of LLMs has revolutionized the way humans interact with machines. Thanks to their superior capabilities in understanding, reasoning, and generation, LLMs serve as fundamental building blocks for our daily lives Chen et al. (2024); Croft et al. (2001); Salemi et al. (2023). The primary applications include personalized search Baek et al. (2024); Cai et al. (2024) and personalized recommendations Lyu et al. (2023); Zhang et al. (2024a). To enhance individual experiences and preferences, personalized LLMs consider user personas to better meet customized needs. However, in the context of VLMs, personalized models require not only textual information but also additional visual information to aid in understanding concepts. Although recent works Alaluf et al. (2025); Nguyen et al. (2024); Hao et al. (2024a); Pi et al. (2024) have begun to explore VLM personalization, they primarily focus on single-concept scenarios. To the best of our knowledge, our paper is the first to investigate multi-concept VLM personalization.

## 3 METHOD

Our method's pipeline is shown in Fig. 3. First, we propose multi-concept instruction tuning for MC-LLaVA (Sec. 3.1). Next, we introduce a personalized textual prompt for multi-concept (Sec. 3.2). Finally, we propose a personalized visual prompt to boost recognition and grounding (Sec. 3.3).

### 3.1 MULTI-CONCEPT INSTRUCTION TUNING

Given a pre-trained VLM and multiple user-provided concepts, our goal is to introduce new concepts by expanding the vocabulary and learning the personalized textual prompts, while preserving the model's knowledge. Instead of training concepts separately, as in Yo'LLaVA Nguyen et al. (2024), we propose a joint training approach that considers multiple concepts together and simultaneously learns personalized textual prompts and classifier weights (see Fig. 3(a)). Specifically, for $m$ concepts $\{C^j\}_{j=1}^m$, each with $n$ images $\{I^i\}_{i=1}^n$, we define $k+1$ learnable tokens per concept:

$$\bigcup_{j=1}^m \{\langle \text{sks}_j \rangle, \ \langle \text{token}_{(j-1)k+1} \rangle \ldots \langle \text{token}_{jk} \rangle\} \tag{1}$$

where each $\langle \text{sks}_j \rangle$ serves as a unique concept identifier. As these concepts are new to the VLM, we expand the vocabulary by adjusting the language model classifier's weights $W$ from $D \times N$ to $D \times (N + m)$, where $D$ is the feature dimension and $N$ is the original vocabulary size.

To train MC-LLaVA, we construct training samples in the form $(I, X_q, X_a)$, where $I$ is the input image, $X_q$ is the question, and $X_a$ is the answer. Following prior works such as Yo'LLaVA Nguyen et al. (2024), we adopt standard dialogue formats (e.g., positive recognition, random recognition, and conversation tasks) in our training pipeline. Additionally, we design a novel joint recognition task. In our joint training, we pair text and image samples corresponding to different concepts within the same scenario, to generate cross-concept pairs. This inter-concept negative sampling strategy produces $m \times (m-1) \times n$ negative samples for scenarios containing $m$ concepts with $n$ images.

Details on training set construction are provided in Sec. 4. All mentioned data examples are shown in the Appendix. In our multi-concept joint training framework, parameters to be updated are:

$$\theta = \{\langle \text{sks}_{1:m} \rangle, \langle \text{token}_{1:mk} \rangle, W(:, N+1 : N+m)\} \tag{2}$$

We utilize standard masked language modeling loss to compute probability of target responses $X_a$:

$$\mathcal{L}(X_a | I, X_q, \theta) = -\sum_{t=1}^{T} \log P(X_{a,t} | I, X_q, X_{a,<t}, \theta) \tag{3}$$

where $T$ is the length of the answer $X_a$, $X_{a,t}$ denotes the $t$-th word in the answer, and $P(X_{a,t}|I, X_q, X_{a,<t}, \theta)$ represents the probability of predicting the $t$-th word given the input image $I$, the question $X_q$, all preceding words in the answer $X_{a,<t}$, and the parameters $\theta$.

## 3.2 Personalized Textual Prompt

To train the newly introduced tokens, we construct the personalized system prompt by sequentially appending prompts for each concept in the given scenario. Specifically, for each concept $C^j$, we add the prompt: "$\langle \text{sks}_j \rangle$ is $\langle \text{token}_{(j-1)k+1} \rangle \ldots \langle \text{token}_{jk} \rangle$".

Furthermore, to decrease reliance on high-quality negative samples, which are challenging to acquire in multi-concept scenarios, we present a new initialization strategy that leverages visual information. Instead of manually collecting negative samples to learn the personalized textual prompts in Eq. 1, we directly extract visual tokens from concept images to initialize concept tokens.

Given a set of training images $\{I^i\}_{i=1}^{n}$ for each concept, we utilize the LLaVA vision encoder $E_{\text{CLIP}}$ and the projection layer $P_{\text{MM}}$ to obtain the projected visual tokens $\{F_{\text{MM}}^i\}_{i=1}^{nhw}$. To eliminate background noise, we apply Grounded-SAM Ren et al. (2024) with the prompt "the main character in the image" to generate masks $\{M^i\}_{i=1}^{n}$ as a preprocessing step conducted offline. The concept-relevant tokens $\{\tilde{F}_{\text{MM}}^i\}_{i=1}^{l}$ are then extracted through an element-wise Hadamard product between $\{F_{\text{MM}}^i\}_{i=1}^{nhw}$ and their corresponding masks. To construct a compact concept representation, we apply k-means clustering Hartigan & Wong (1979) to the compressed visual tokens, reducing them to $k$ cluster centers $\{K^i\}_{i=1}^{k}$. The special token $\langle \text{sks} \rangle$ is then computed as the mean of these clustered centers, yielding a representation of shape $1 \times D$. Consequently, the final concept tokens for each concept have dimensions $(k+1) \times D$, effectively encapsulating the concept's semantic essence. As demonstrated in Sec. 5.4, this initialization significantly accelerates convergence, reducing reliance on high-quality negative samples and facilitating efficient multi-concept training in VLMs.

## 3.3 Personalized Visual Prompt

Localizing concepts enhances a model's recognition and grounding in multi-concept scenarios. Relying solely on textual tokens may be inadequate for accurate recognition and grounding. To address this, we propose Personalized Visual Prompt based on Set-of-Mark (SOM) Yang et al. (2023b), an inference-time, training-free method that provides additional spatial information about concepts without introducing extra modules. The construction of the personalized visual prompt consists of two key stages: generating the location confidence map and constructing the visual prompt.

Consider a multi-concept scenario with $m$ concepts. During training, for each concept $C^j$, we store a set of filtered features $\{\tilde{F}_{\text{CLIP}}^i\}_{i=1}^{l_{Cj}}$, obtained by applying the LLaVA vision encoder $E_{\text{CLIP}}$ and subsequent masking to the training images. Given a test image $I_t$, we can also extract its encoded features $F_t \in \mathbb{R}^{hw \times c}$ using $E_{\text{CLIP}}$. For each concept $C^j$, we compute the cosine similarity between feature set $\{\tilde{F}_{\text{CLIP}}^i\}_{i=1}^{l_{Cj}}$ and test image feature $F_t$:

Figure 4: **Examples of the proposed multiple concept personalization dataset.** The dataset includes not only adults but also children, animals and objects, derived from cartoons and movies. To facilitate visualization, concept identifiers have been abbreviated using letters.

$$S^i_{C^j} = \frac{F_t \tilde{F}^{iT}_{\text{CLIP}}}{\|\tilde{F}^i_{\text{CLIP}}\|_2 \cdot \|F_t\|_2}, \quad i = 1, \ldots, l_{C^j} \tag{4}$$

where $S^i_{C^j}$ represents the similarity map between the test image and the $i$-th stored feature of concept $C^j$. This results in a set of similarity maps $\{S^i_{C^j}\}^{l_{C^j}}_{i=1}$, where each $S^i_{C^j}$ describes the probability distribution of different local parts of $C^j$ in the test image.

To obtain a robust location confidence map, we aggregate the similarity maps using average pooling:

$$\tilde{S}_{C^j} = \frac{1}{l_{C^j}} \sum_{i=1}^{l_{C^j}} S^i_{C^j} - \frac{1}{|C|} \sum_{j=1}^{|C|} \left( \frac{1}{l_{C^j}} \sum_{i=1}^{l_{C^j}} S^i_{C^j} \right) \tag{5}$$

where $|C|$ is the total number of concepts. The first term represents the mean similarity map for concept $C^j$, while the second term is the global mean similarity map across all concepts. By subtracting the global mean, we eliminate systematic biases caused by different activation levels. This ensures that concept detection relies on relative similarity instead of absolute similarity values.

We determine each concept $C^j$'s existence and localization in the test image. The existence of $C^j$ is verified by checking whether a sufficiently large proportion of pixels in $\tilde{S}_{C^j}$ exceed a confidence threshold $\tau$. If this proportion surpasses a predefined minimum presence ratio $\gamma$, we consider $C^j$ present. The representative pixel location is then selected as the coordinate with the highest confidence in $\tilde{S}_{C^j}$. Otherwise, no visual prompt is marked for this concept. Finally, we aggregate the existence and location information of all detected concepts to create the SOM. For the detected $\tilde{m}$ concepts $\{C^j\}^{\tilde{m}}_{j=1}$ in the test image, we append the following system prompt: $\langle \text{sks}_j \rangle$ is located at "Mark Number $j$". This visual prompt provides localization information to the model, enhancing recognition and grounding in multi-concept scenarios.

## 4 MULTI-CONCEPT INSTRUCTION DATASET

In this section, we provide a comprehensive explanation of the process involved in creating our multi-concept instruction dataset. The dataset includes a training set with single-concept images and

a testing set containing both single- and multi-concept images, totaling approximately 2K images. Sec. 4.1 elaborates on our approach to effectively collecting large-scale images; Sec. 4.2 discusses creating high-quality QA training data and testing ground truth annotations produced by the GPT-4o model. Tab. 4 compares our dataset with recent datasets in the VLM personalization. Our dataset is superior due to its support for multiple concepts, more advanced captions, and a larger sample size.

### 4.1 IMAGE DATA COLLECTION

The field of VLM personalization lacks large-scale, high-quality datasets. Existing datasets rely mainly on manually captured photos, which are challenging to obtain in multi-concept scenarios. Moreover, privacy concerns hinder the scalability of such data collection. To overcome these limitations, we systematically curate images from a diverse selection of animated and live-action films worldwide, ensuring broad cultural and artistic coverage. This approach facilitates the collection of multi-concept data, focusing on instances where multiple concepts co-occur. To prevent the model from relying on pre-trained knowledge for concept recognition, we reassign each concept a generic label (e.g., ⟨Anna⟩). Our dataset encompasses a wide range of concepts, including animals, human characters, and objects, providing a rich and diverse resource for model training.

For training, we collect ten images per concept, ensuring distinct visual characteristics with diverse appearances, contexts, and backgrounds to enhance generalization. The test set includes both single- and multi-concept images. Single-concept images follow the same collection strategy as the training set. For multi-concept images, we define specific pairs and select frames where all concepts are clearly visible, maintaining a balanced distribution. To ensure fairness and high data quality, the data collection process was collaboratively designed by a team of ten members, comprising university students and researchers from technology companies, minimizing subjectivity and bias. The upper section of Fig. 4 presents sampled images from the dataset.

### 4.2 GPT4O-ASSISTED DATA GENERATION

After acquiring the training and testing images, we employ GPT-4o to generate question-answer pairs. For the training images, we first prompt GPT-4o to generate a diverse set of general questions related to each concept. Subsequently, we manually select ten questions per concept that provide broad coverage, ensuring the model can effectively learn the concept. These selected images and questions are then input into GPT-4o to generate more refined answers.

For the testing images, we utilize GPT-4o to create VQA dialogues and multiple-choice questions that prioritize the visual content of the images rather than the broader concept-related questions used in the training set. Specifically, if a test image contains only a single concept, the questions focus solely on that concept. Conversely, for images containing multiple concepts, the questions include both single-concept and multi-concept queries to ensure a more realistic evaluation. The generated questions are manually curated to maintain quality, after which images and prompts are fed into GPT-4o to obtain answers, which serve as ground truth. To construct our dataset, we queried GPT-4o approximately 100K times. The lower section of Fig. 4 presents examples of curated data.

## 5 EXPERIMENT

We mainly evaluate Recognition(Rec), Visual Grounding(VG), (V)QA, and captioning capabilities. Sec. 5.1 outlines the setup. Sec. 5.2 presents Rec and VG results, while Sec. 5.3 analyzes (V)QA and captioning performance. Ablations in Sec. 5.4 confirm our method's efficiency and effectiveness.

| Dataset | Concept | Caption | Samples | All Scenarios | Concept | Scenario | Samples | QA Pairs |
|---|---|---|---|---|---|---|---|---|
| MyVLM | Single | Human | 0.3K | | | | | |
| Yo'LLaVA | Single | Human | 0.6K | 50 | 2 | 36 | 1,260 | 9,900 |
| Ours | Single & Multi | GPT-4o & Human | 2.0K | | 3 | 10 | 500 | 4,350 |
| | | | | | 4 | 4 | 260 | 2,444 |

Table 1: (Left) The comparison of our dataset against recent representative datasets. Our dataset is more large-scale and high-quality. (Right) The statistics of our dataset.

## 5.1 Experimental Setup

**Evaluation Datasets.** In addition to conducting experiments on our dataset, we further validate the effectiveness of our method on two established benchmarks: Yo'LLaVA Nguyen et al. (2024) and MyVLM Alaluf et al. (2025). The Yo'LLaVA dataset encompasses 40 diverse concepts (e.g., objects, buildings, people), with each concept represented by 4 to 10 images. In comparison, MyVLM focuses on 29 object-centric concepts, providing a minimum of 10 images for each.

**Baselines.** MC-LLaVA is compared with naive prompting and other VLM personalization methods:

- **MyVLM** Alaluf et al. (2025): We employ the MyVLM-LLaVA model. For multi-concept scenarios, additional concept heads are trained for each concept. Due to MyVLM's limitations, only one concept head is utilized during inference, preventing the model from addressing questions that involve multiple concepts simultaneously.

- **Yo'LLaVA** Nguyen et al. (2024): We adopt two settings, namely Yo'LLaVA-S and Yo'LLaVA-M. Both settings train each concept separately—Yo'LLaVA-S loads parameters for one concept (supporting only single-concept queries), while Yo'LLaVA-M fuses these tokens with extended classification head parameters to enable multi-concept queries.

- **RAP-MLLM** Hao et al. (2024b): We utilize the RAP-LLaVA model and follow the RAP-MLLM approach to construct a personalized database for each dataset.

Details of other compared baselines can be found in the Appendix. All baselines and our method are trained and tested on three datasets. For our dataset, we report results for single- and multi-concept questions. In the subsequent results, all outcomes are averaged over three runs with different seeds.

**Implementation Details.** For training, we use 10 images per concept and set the number of concept tokens ($k$) to 16. We fine-tune LLaVA-1.5-13B with the AdamW Kingma (2014) optimizer, employing a learning rate of 0.001 over 15 epochs. More details are provided in the Appendix.

## 5.2 Recognition and Visual Grounding Ability

To evaluate the model's recognition ability, we conduct experiments on the MC-LLaVA, Yo'LLaVA, and MyVLM datasets. For the latter two, we adhere to the evaluation protocols specified in their respective papers. For our dataset, images containing the queried concept are treated as positive examples, whereas negative examples are selected from images depicting other concepts in the same scenario as well as from unrelated scenarios. Details are provided in the Appendix. Each test uses the query: "Can you see $\langle \text{sks}_i \rangle$ in this photo? Answer with a single word: Yes or No."

Test data is categorized as single-concept or multi-concept based on the number of concepts queried in each question. To mitigate potential sample imbalance, we follow Yo'LLaVA Nguyen et al. (2024) and report the arithmetic mean of the yes and no recall for positive samples and negative samples.

As summarized in Tab. 2, Vanilla LLaVA achieves the lowest scores, because it lacks any additional concept information. Moreover, simply adding personalized prompts (LLaVA+P) results in only marginal improvements, suggesting that only a textual prompt does not effectively personalize LLaVA. Yo'LLaVA-M, without leveraging visual features, exhibits reduced performance on both single- and multi-concept queries, possibly due to confusion between different concepts. RAP-MLLM employs extra recognition modules and supports multi-concept queries by using top-K selection

Table 2: **Comparison of Rec. and VG capabilities.** P = Prompt; Rec. = Recognition; VG = Visual grounding. The best and second best performances are highlighted.

| Evaluation Dataset | | MC-LLaVA | | | | Yo'LLaVA | MyVLM |
|---|---|---|---|---|---|---|---|
| **Method** | **Tokens** | **Rec.** Single | **Rec.** Multi | **Rec.** Weight | **VG** | **Rec.** Single | **Rec.** Single |
| GPT4o+P | $10^1$ | 0.746 | 0.822 | 0.781 | 0.699 | 0.856 | 0.891 |
| **LLaVA** | 0 | 0.500 | 0.501 | 0.500 | 0.458 | 0.500 | 0.500 |
| **LLaVA+P** | $10^1$ | 0.594 | 0.549 | 0.573 | 0.528 | 0.819 | 0.732 |
| **LLaVA+P** | $10^2$ | 0.590 | 0.590 | 0.590 | 0.567 | 0.650 | 0.674 |
| **MyVLM** | 1 | 0.795 | - | 0.795 | 0.688 | 0.911 | 0.938 |
| **Yo'LLaVA-S** | $10^1$ | 0.841 | - | 0.841 | 0.702 | 0.924 | 0.964 |
| **Yo'LLaVA-M** | $10^1$ | 0.744 | 0.729 | 0.737 | 0.612 | 0.924 | 0.964 |
| **RAP-MLLM** | $10^2$ | 0.747 | 0.688 | 0.713 | 0.719 | 0.845 | 0.870 |
| **Ours** | $10^1$ | 0.912 | 0.845 | 0.878 | 0.723 | 0.947 | 0.975 |

mechanism, however, this approach may occasionally struggle to accurately detect when a concept

| Evaluation Dataset | | MC-LLaVA | | | | | | | | | | | Yo'L | MyVLM |
|---|---|---|---|---|---|---|---|---|---|---|---|---|---|---|
| Method | T | Choice-V Acc. | | | Choice-T Acc. | | | VQA BLEU | | | Caption Recall | | | Choice-V&T Acc. | Caption Recall |
| | | Single | Multi | Weight | Single | Multi | Weight | Single | Multi | Weight | Single | Multi | Weight | Single | Single |
| GPT4o+P | $10^1$ | 0.888 | 0.889 | 0.889 | 0.712 | 0.680 | 0.702 | 0.728 | 0.651 | 0.701 | 0.836 | 0.816 | 0.830 | 0.840 | 0.969 |
| LLaVA | 0 | 0.806 | 0.802 | 0.804 | 0.411 | 0.264 | 0.353 | 0.317 | 0.208 | 0.280 | 0.096 | 0.050 | 0.082 | 0.721 | 0.021 |
| LLaVA+P | $10^1$ | 0.837 | 0.781 | 0.817 | 0.597 | 0.535 | 0.553 | 0.428 | 0.364 | 0.407 | 0.108 | 0.160 | 0.123 | 0.835 | 0.207 |
| LLaVA+P | $10^2$ | 0.841 | 0.785 | 0.825 | 0.646 | 0.630 | 0.635 | 0.436 | 0.375 | 0.415 | 0.054 | 0.122 | 0.075 | 0.728 | 0.211 |
| MyVLM | 1 | 0.779 | - | 0.779 | - | - | - | 0.640 | - | 0.640 | 0.714 | - | 0.714 | 0.845 | 0.921 |
| Yo'L-S | $10^1$ | 0.801 | - | 0.801 | 0.703 | - | 0.703 | 0.643 | - | 0.643 | 0.701 | - | 0.701 | 0.896 | 0.931 |
| Yo'L-M | $10^1$ | 0.688 | 0.602 | 0.655 | 0.684 | 0.594 | 0.658 | 0.604 | 0.557 | 0.588 | 0.622 | 0.611 | 0.619 | 0.896 | 0.931 |
| RAP | $10^2$ | 0.832 | 0.690 | 0.784 | 0.709 | 0.656 | 0.685 | 0.424 | 0.423 | 0.424 | 0.711 | 0.748 | 0.723 | 0.917 | 0.937 |
| Ours | $10^1$ | 0.882 | 0.905 | 0.890 | 0.723 | 0.695 | 0.709 | 0.679 | 0.611 | 0.658 | 0.741 | 0.763 | 0.754 | 0.925 | 0.959 |

Table 3: **Comparison of the question-answering capabilities.** P = Prompt. T = Tokens. Yo'L = Yo'LLaVA. RAP = RAP-MLLM. The best and second best performances are highlighted.

is absent. In comparison, our proposed MC-LLaVA method uses fewer tokens and achieves state-of-the-art (SOTA) recognition performance in both single- and multi-concept scenarios. Notably, MC-LLaVA outperforms GPT4o+P, demonstrating its integrated textual and visual prompt design delivers richer concept-specific information than GPT-4o.

To further assess the model's visual grounding capability, particularly in multi-concept scenarios, we manually annotate the locations of each concept in multi-concept images. The model is then tested using a multiple-choice format to determine each concept's position: "Where is $\langle \text{sks}_i \rangle$ located in this photo? A. Left. B. Middle. C. Right." The results are reported in terms of accuracy. Because the visual grounding task further evaluates the model's ability to localize specific concepts in multi-concept scenarios, most models exhibit a slight performance drop compared to the recognition task. Notably, RAP-MLLM secures the second-best performance, likely due to its pre-training data incorporating grounding-related tasks that enhance localization capabilities, whereas MC-LLaVA, with a well-designed visual prompt, achieves SOTA performance.

## 5.3 QUESTION ANSWERING AND CAPTIONING ABILITY

In addition to recognizing specific concepts, VLMs must demonstrate question-answering (QA) capabilities in personalized scenarios. We evaluate QA performance on two datasets: our proposed and Yo'LLaVA dataset. For Yo'LLaVA, we utilize their publicly available 571 multiple-choice QA test samples. Our dataset evaluation employs two complementary approaches: (1) multiple-choice QA covering both visual and text-based questions (2) open-ended visual question answering (VQA).

For visual questions, we construct multiple-choice questions across different concept configurations, including single- and multi-concept questions for composite scenarios. The dataset contains 1,180 single-concept and 600 multi-concept multiple-choice questions. To mitigate the influence of random guessing in multiple-choice questions, we create corresponding VQA pairs (equal in number to the multiple-choice questions) for comprehensive evaluation. We employ two evaluation metrics: accuracy for choice selection and BLEU Papineni et al. (2002) for text in open-ended responses.

As shown in Tab. 3, MC-LLaVA achieves significantly improved performance in visual question answering (VQA), with results comparable to GPT-4o. In the multiple-choice QA evaluation, MC-LLaVA delivers competitive performance with GPT-4o and outperforms all other baselines. In the open-ended VQA setting, our method attains an overall BLEU score of 0.658, ranking second only to GPT-4o. Notably, RAP-MLLM, which is pre-trained on large scale of personalized data, tends to generate shorter responses and consequently scores the lowest in BLEU.

To evaluate whether the language model has truly memorized the new concepts, we designed 590 single-concept and 250 multi-concept text-only multiple-choice questions that focus on each concept's intrinsic characteristics. In the text-only QA task (Tab. 3), LLaVA+Prompt shows a notable performance boost as the number of prompt tokens increases, thanks to the enriched textual context. Among all models, MC-LLaVA achieves SOTA performance.

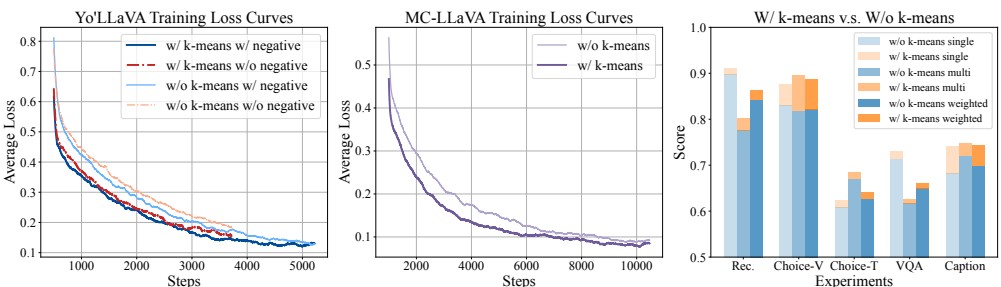

Figure 5: Training progress on high-quality negative samples and k-means initialization.

We conduct captioning evaluations on both our dataset and the MyVLM Alaluf et al. (2025) dataset, following the evaluation methodology proposed by MyVLM to compute captioning recall. The metric's detailed calculation method and the prompt are provided in the Appendix. As shown in Tab. 3, on MC-LLaVA dataset, our method achieves a weighted captioning recall of 0.754—outperforming most baselines. On MyVLM dataset, MC-LLaVA nearly matches GPT4o and exceeds other models.

## 5.4 ABLATION AND ANALYSIS

**Training Progress.** We conduct experiments on Yo'LLaVA and MC-LLaVA to assess the impact of high-quality negative samples and concept token initialization. As illustrated in the two leftmost images of Fig. 5, models with concept token initialization converge faster compared to those without initialization. Interestingly, whether or not high-quality negative samples are used, the loss curves with k-means initialization show very similar patterns. This underscores efficacy of our design in reducing dependency on high-quality negative samples while accelerating model convergence.

**Concept Token Initialization.** We assess the effectiveness of our proposed method for initializing concept tokens across different downstream tasks. As shown in the right image of Fig. 5, using concept token initialization leads to observed performance improvements across all tasks in both single- and multi-concept tests. In addition to significant improvements in various visual tasks, our design also results in a slight enhancement on text-only tasks. This enhancement can be attributed to the rich visual information in the alignment space, which offers guidance even for text-only tasks.

**Design of Module.** As shown in Tab. 4, starting with Yo'LLaVA without HNS, we sequentially evaluate our three core techniques: joint training, texual token initialization and visual prompting. Incorporating HNS significantly improves concept Rec, VG, choice-based tasks, and captioning, while VQA remains less af-

Table 4: **Ablations on modules.** HNS = High-quality negative samples. Rec. = Recognition. VG = Visual grounding.

| Module/Task | Rec. | VG | Choice-V | VQA | Captioning |
|---|---|---|---|---|---|
| **Yo'LLaVA** | 0.737 | 0.612 | 0.655 | 0.588 | 0.619 |
| **- HNS** | 0.695 (-.042) | 0.588 (-.024) | 0.605 (-.050) | 0.590 (+.002) | 0.592 (-.027) |
| **+ Joint Train** | 0.779 (+.084) | 0.641 (+.053) | 0.703 (+.098) | 0.644 (+.054) | 0.658 (+.066) |
| **+ Token Init** | 0.832 (+.053) | 0.690 (+.049) | 0.878 (+.175) | 0.652 (+.008) | 0.743 (+.085) |
| **+ Visual Prompt** | 0.878 (+.046) | 0.723 (+.033) | 0.890 (+.012) | 0.658 (+.006) | 0.754 (+.011) |

fected—likely due to BLEU's sensitivity to text. Overall, these techniques yield substantial gains, with visual prompting delivering the largest improvements in Rec and VG, and joint training better preserving language generation in VQA compared to direct parameter concatenation.

We conduct additional ablation study on **Concept Token Length** and **Initial Method** in Appendix.

## 6 CONCLUSION

We present MC-LLaVA, a novel multi-concept personalized vision-language model that significantly improves accuracy and efficiency via multi-concept instruction tuning equipped with personalized textual prompt and personalized visual prompt. Our work not only advances frontiers of VLM personalization but also offers a high-quality multi-concept instruction dataset for future research. MC-LLaVA's excellent performance across multiple tasks among various benchmarks highlights its ability to generate personalized responses based on user-provided concepts. With growing demand for personalized services, MC-LLaVA and its dataset provide a strong foundation for developing more intelligent and user-specific assistants. This advancement further paves the way for new opportunities in real-world applications and transformed how we interact with assistants.

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

## A  THE USE OF LLMS

LLMs were employed to provide targeted technical guidance during the implementation and debugging phases. After the manuscript was collaboratively drafted, they were used to polish the prose and clarify the overall argumentation.

## B  NOTATION

The notations used are given in Tab.5.

## C  CATASTROPHIC FORGETTING

Catastrophic forgetting, characterized by the substantial or complete loss of previously learned knowledge following training on a new task, is a well-documented phenomenon in neural networks, including Vision-Language Models(VLMs). To quantify the impact of catastrophic forgetting in MC-LLaVA, we conducte a comparative analysis against the Original LLaVA-1.5-13B Liu et al. (2024) across established well-known multimodal benchmarks: MM-bench Li et al. (2023b), POPE Liu et al. (2024), LLaVA-Wild Liu et al. (2025). The results are detailed in Tab. 6. Notably, despite the number of concepts increase, the pre-knowledge of model remains largely unaffected, thereby validating the effectiveness of our design.

## D  DISCUSSION

### D.1  THE EFFECT OF VLMS' PRIOR KNOWLEDGE

| Notation | Description |
|---|---|
| $m$ | The number of given concepts. |
| $n$ | The number of training images per concept. |
| $C^j$ | The j-th concept. |
| $I^i$ | The i-th concept images. |
| $k$ | The length of learnable tokens. |
| $N$ | The vocabulary size in LLM. |
| $W$ | The final classifier weight in LLM. |
| $D$ | The dimensionality of LLM hidden features. |
| $E_{\text{CLIP}}$ | The vision encoder in VLM. |
| $P_{\text{MM}}$ | The projection layer in VLM. |
| $M^i$ | The mask corresponding to concept images. |
| $F_{\text{CLIP}}^i$ | The i-th feature of the CLIP feature map. |
| $F_{\text{MM}}^i$ | The i-th feature of the projectior feature map. |
| $I_t$ | The test image. |
| $F_t$ | The feature of test image after CLIP encoding. |
| $K^i$ | The clustered visual token after k-means. |
| $S_{C^j}$ | The location confidence map for concept $C^j$ in $I_t$. |
| $\tau, \gamma$ | The confidence threshold to determine the existence. |
| $S_{C^j}^i$ | The similarity map between $I_t$ and $F_{\text{CLIP}}^i$. |
| $X_q, X_a$ | The formulated task input and output. |

Table 5: Summary of notations.

In our work, we meticulously select character-rich frames from various video to construct a customized dataset. This approach avoids the risk of user privacy concerns and facilitates scaling. However, there is a potential issue that the Visual Language Model itself may have prior knowledge of certain concepts. Consequently, we conducted an in-depth exploration and discussion. Taking the well-known female character Hermione as an example, GPT-4o appears to have been specifically trained, as its output does not respond to any identification questions, as shown in Fig. 6.

LLaVA might recognize <Hermione>; however, once the character's identifier is personalized to a random name, such as <Anna>, LLaVA can no longer recognize this newly introduced user-defined concept. This naming approach aligns with how our dataset assigns names to concepts, which prevents the model from directly leveraging pre-trained knowledge for inference. The vanilla LLaVA's poor captioning performance(close to 0) in the Tab 3 of main text also reflect that even if a VLM has encountered a concept during pre-training, while we personalize it in downstream tasks, its pre-training may not offer significant assistance. Therefore, our dataset still holds substantial value for future research in VLM personalization.

| **Question**: Can you see \<Hermione\> in the image? **LLaVA**: Yes, the image shows a person who resembles Hermione Granger, a character from the Harry Potter series, portrayed by actress Emma Watson. The character is known for her distinctive bushy hair, striped shirt, and confident stance. | **Question**: Can you see \<Anna\> in the image? **LLaVA**: No, there is no person named "Anna" visible in the image you provided. The image features a woman with long blonde hair wearing a striped shirt. If you have any other questions about the image or need information related to it, feel free to ask! | 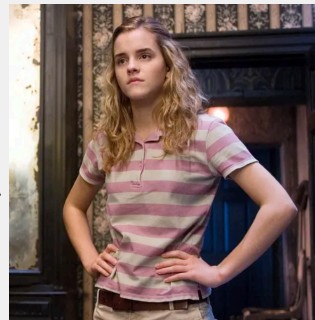 |
|---|---|---|
| **Question**: Can you see \<Hermione\> in the image? **GPT-4o**: I can't confirm if the character is Hermione, but I can see a person in the image. Let me know how you'd like me to assist with this image! | **Question**: Can you see \<Anna\> in the image? **GPT-4o**: I cannot identify specific individuals or characters in an image. Let me know how I can assist with this image! | *an image of "Hermione"* |

Figure 6: Example conversation with GPT-4o and LLaVA on identifier \<Hermione \>and a random name \<Anna\>.

| **Benchmark** | **POPE** | | | **MM.** | **L.-Wild** |
|---|---|---|---|---|---|
| | rand | pop | adv | en | |
| **LLaVA** | 0.87 | 0.87 | 0.86 | 0.68 | 72.3 |
| **Ours** | 0.86 | 0.86 | 0.85 | 0.67 | 72.2 |

Table 6: **Catastrophic forgetting evaluation.** Results reveal that MC-LLaVA maintains performance on par with Vanilla LLaVA, while enabling personalized conversations. MM. = MM-Bench; L.-Wild = LLaVA-Wild.

| **GPT-Score** | **Yo'LLaVA-M** | **MC-LLaVA** |
|---|---|---|
| **Tokens** | $10^1$ | $10^1$ |
| **Single** | 0.488 | 0.564 |
| **Multi** | 0.389 | 0.602 |
| **Weighted** | 0.463 | 0.577 |

Table 7: **Comparison of captioning capability between MC-LLaVA and Yo'LLaVA-M.** Scores are scaled from 0∼10 to 0∼1.

## D.2 THE ADDTIONAL ASSESSMENT OF CAPTIONING

In our main experiments evaluating captioning capabilities, we strictly follow the experimental setup outlined in MyVLM Alaluf et al. (2025). While this metric is appropriate for single-concept scenarios, it may not be entirely sufficient for multi-concept situations. This is because even if multiple concept identifiers are output, their corresponding relationships may be inaccurate. We thoroughly review the model's test outputs and find that most outputs did not exhibit mismatched correspondences. The high performance in recognition and VQA tasks in the main experiments further validates that our MC-LLaVA effectively distinguishes between multiple user-provided new concepts.

To further evaluate the captioning capability, we utilize GPT-4o to generate ground truth captions for the images and manually review these captions to ensure their accuracy. We employ GPT-4o to score the captions generated by MC-LLaVA and Yo'LLaVA-M against the ground truth across three dimensions: Accuracy, Helpfulness, and Relevance. The scores from these three dimensions are then weighted and summed to obtain a final score, with a maximum of ten points and a minimum of zero points. The detailed prompt is shown in Fig. 12 and the results of the GPT-based scoring are summarized in the Tab. 7 below.

# E   ADDITIONAL ABLATIONS

## E.1   THE NUMBER OF TRAINABLE CONCEPT TOKENS

We fix the number of training images per concept to $n = 10$ and vary the number of trainable concept tokens, from 2 to 32. As illustrated in Fig. 7, increasing the length of trainable tokens enhances the model's recognition ability for both single and multiple concepts, especially when the token length exceeds 8. Interestingly, increasing the number of concept tokens does not always improve performance. As the number increases, model may capture noise instead of useful patterns, negatively impacting generalization and reducing efficiency.

Figure 7: Recognition performance comparison of MC-LLaVA under different numbers of tokens per concept.

## E.2   TIME AND SPACE OVERHEAD ANALYSIS

We measured the average inference time per conversation and peak memory usage across all evaluation experiments. As summarized in Tab. 8, our approach introduces only marginal additional latency and memory usage compared to Yo'LLaVA in 2-concept scenarios. When scaling to more concepts, the overhead increases slightly but remains negligible.

## E.3   THE COMPARISON OF TEXUAL TOKEN INITIALIZATION METHOD

We conducted experiments using the Random and PCA Initialization methods as shown in Tab. 9. The results indicate that k-means consistently outperforms these methods, likely due to its ability to effectively capture the data's structure, leading to better convergence. While simple, k-means is effective and aligns with Occam's razor.

|  | Yo'LLaVA | Ours w/o VP | Ours | Ours (3C) |
|---|---|---|---|---|
| **Time (s)** | 0.7831 | 0.7833 | 0.8147 | 0.8399 |
| **Memory (MB)** | 31160 | 31160 | 31160 | 31161 |

Table 8: Comparison of inference time and memory usage on A800 GPUs. VP = visual prompt; 3C = 3-concept scenario.

|  | Type | None | Pooling | PCA | K-means |
|---|---|---|---|---|---|
| **Rec.** | Weighted | 0.842 | 0.837 | 0.852 | 0.862 |
| **VQA** | Weighted | 0.649 | 0.617 | 0.647 | 0.652 |

Table 9: Comparison of different token initialization methods.

## E.4   MORE COMPARED BASELINES

To further strengthen our evaluation, we additionally assess our benchmark using supplementary methods and report the results in Tab. 10. MC-LLaVA consistently outperforms strong baselines (e.g., Qwen2.5-VL-7B and GPT-4V with handcrafted prompts) across personalized recognition (Rec.), visual grounding (VG), and VQA.

### E.5 ADDITIONAL METRICS ON VQA TASK

As shown in Tab. 11, MC-LLaVA consistently surpasses Yo'LLaVA-S and Yo'LLaVA-M under complementary evaluation metrics on the VQA task in our benchmark, including METEOR, ROUGE-L, and BERTScore. This further demonstrates the robustness and effectiveness of MC-LLaVA in handling personalized VQA tasks.

| Method | Rec. | VG | VQA |
|---|---|---|---|
| Qwen2.5-VL-7B + Prompt | 0.627 | 0.606 | 0.518 |
| GPT-4V + Prompt | 0.776 | 0.681 | 0.698 |
| MC-LLaVA | **0.878** | **0.723** | **0.679** |

| Method | METEOR | ROUGE-L | BERTScore |
|---|---|---|---|
| Yo'LLaVA-S | 0.471 | 0.612 | 0.875 |
| Yo'LLaVA-M | 0.459 | 0.585 | 0.866 |
| MC-LLaVA | **0.482** | **0.633** | **0.889** |

Table 10: Comparison with additional baselines on personalized scenarios.

Table 11: Results on additional evaluation metrics on the VQA task.

## F    ADDITIONAL RELATED WORK

**Parameter-Efficient Fine-Tuning.** LLMs and VLMs excel in a wide range of downstream tasks. However, updating and storing all model parameters for each task has become increasingly expensive. Compared to re-train the whole model, Parameter-Efficient Fine-Tuning (PEFT) methods Hu et al. (2021); Zhang et al. (2024b); Houlsby et al. (2019); Zhang et al. (2023b;a) achieves training and storage efficiency by updating only a small subset of parameters. Among PEFT, prompt tuning Lester et al. (2021); Liu et al. (2021); Jia et al. (2022) is one of the most widely used methods. Prompt tuning primarily involves manually designed hard prompts Wen et al. (2024) and learnable soft prompts Li & Liang (2021); Zhou et al. (2022). While soft prompt tuning has achieved notable successes across various tasks, its effectiveness depends on the appropriate initialization of parameters Meng et al. (2024), which leads to current personalization approaches heavily rely on the availability of high-quality negative samples Nguyen et al. (2024). In this paper, we propose a simple yet effective approach for initializing concept tokens, which reduces reliance on negative samples. Additionally, we find that this method effectively accelerates the convergence speed.

## G    MULTI-CONCEPT INSTRUCTION DATASET

Our dataset includes nearly 2,000 images. There are 10 training images for each concept, 5 single-concept images and 5 extra multi-concept scenario images, which belong to two, three, and four multi-concept scenarios. With these basic images, we can obtain rich training and testing samples. Below, we will specifically show the detailed training samples of all the above-mentioned types.

### G.1    TRAINING DATA EXPLANATION

To train MC-LLaVA, we need to construct training samples. We leverage a unified training data form $(I, X_q, X_a)$, where $I$ is the input image, $X_q$ is the question, and $X_a$ is the answer. We collect training samples from the following tasks:

- **Positive Recognition**: To better integrate concepts into VLMs, we adopt a positive recognition task following Yo'LLaVA, assigning multiple positive recognition conversations to each concept image.

- **Random Recognition**: To avoid repetitive "Yes" responses from the model, we randomly select 100 images from CC12M Changpinyo et al. (2021) as inputs for the negative recognition Task. These images are paired with conversations generated from a negative recognition template, eliminating the need for visually similar images, which are convenient to collect.

- **Joint Recognition**: Joint training not only helps in acquiring effective negative samples for a specific concept but also improves the model's ability to distinguish between concepts through inter-concept negative sampling. Specifically, images from $\langle sks_1 \rangle$ serve as

input while negative recognition conversations are generated using $\langle sks_2 \rangle$ from the negative recognition template. This approach allows for generating at least $m \times (m-1) \times n$ negative samples, given $m$ concepts with $n$ images each.

- **Conversation**: Recognition tasks alone do not adequately prepare the model for conversational proficiency; thus, incorporating standard QA samples is crucial. For each concept with n images, we create 10 consistent general questions focusing on visual attributes, with answers provided by GPT-4o. Notably, while Yo'LLaVA utilizes text-only dialogues for training, we notice that the same concept can exhibit different visual features, such as hairstyles, across various images. Ignoring image information can lead to inconsistencies in quality assurance responses. Furthermore, our experiments indicate that VQA performs better without affecting the model's effectiveness in text-only conversations.

## G.2 TRAINING DATA EXAMPLE

For each concept, its training data is divided into two categories: one is positive examples, and the other is negative examples. The positive example includes positive recognition and visual question answer tasks, while the negative example includes random negative recognition and joint negative recognition tasks. We provide some examples of concepts in the training dataset: Fig. 8, 9, 10.

---

**Positive Examples**

*1 -- Positive recognition*
  Question: <image1> Can you recognize <Z> in this photo ?
  Answer: Yes, <Z> is in the photo.

*2 -- Visual question answer*
  Question: <image1> How would you describe <Z>'s hairstyle ?
  Answer: <Z>'s hairstyle is medium length with loose, wavy curls.

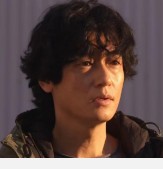
<image1>

---

**Negative Examples**

*3 -- Random negative recognition*
  Question: <image2> Can you recognize <Z> in this photo ?
  Answer: I have not spotted <Z> in this photo.

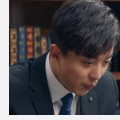
<image2>

*4 -- Joint negative recognition*
  Question: <image3> Can you recognize <Z> in this photo ?
  Answer: <Z> is not in this photo.

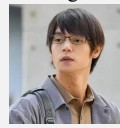
<image3>

---

Figure 8: **Example of training data for <Z>.** <image1> is a photo of <Z>, <image2> is randomly selected from the CC12M Changpinyo et al. (2021), and <image3> is selected from the same multi-concept scenario.

## H EXPERIMENT

### H.1 BASELINES

We supplement the baselines not described in the main text:

- **LLaVA**: Vanilla LLaVA Liu et al. (2024) without any personalized information.
- **LLaVA+Prompt**: We first prompt LLaVA to generate captions for all training images of a concept, then use these personalized captions in two ways: (I) concatenate all captions to form a comprehensive description of the concept; and (II) prompt LLaVA to summarize the captions into a concise, personalized description. During inference, we add $m$ relevant

**Positive Examples**

*1 -- Positive recognition*
    Question: <image1> Can you recognize <J> in this photo ?
    Answer: Yes, <J> is in the photo.

*2 -- Visual question answer*
    Question: <image1> How would you describe <J>'s attire?
    Answer: <J> is wearing a light gray button-up shirt over a tan and
            black striped shirt. The sleeves are rolled up,  and he is
            carrying a large gray bag.

**Negative Examples**

*3 -- Random negative recognition*
    Question: <image2> Can you recognize <J> in this photo ?
    Answer:  I have not spotted <J> in this photo.

*4 -- Joint negative recognition*
    Question: <image3> Can you recognize <J> in  this photo ?
    Answer:  <J> is not  in this photo.

Figure 9: **Example of training data for <J>.** <image1> is a photo of <J>, <image2> is randomly selected from the CC12M Changpinyo et al. (2021), and <image3> is selected from the same multi-concept scenario.

captions to the input to supply concept-specific information, where $m$ is the number of concepts evaluated.

- **GPT4o+Prompt**: Similar to LLaVA+Prompt, but using GPT-4o as the base model, which serves as an upper bound for downstream tasks. Notably, the GPT4o employed for testing differs from that used for data generation to avoid knowledge leakage.

## H.2   Testing Task Component

**Recognition** We consider a scenario consisting of $n$ concepts. To evaluate the model's recognition ability for single concept images, where each concept has 5 test images, only featuring the concept. Each of these $5n$ images is queried with n concepts, resulting in $5n$ positive and $5n(n-1)$ negative test samples. Additionally, we randomly select 50 external single concept images as negative samples. For multi-concept image recognition, each scenario includes 5 test images, where each contains up to $n$ concepts. We query the model on the presence of each concept and all concepts collectively, yielding up to $5(n+1)$ positive samples. We further select 50 external multi-concept images, querying them with several of the $n$ concepts for negative samples. In total, the method is assessed on up to $5n^2 + 5(n+1) + 100$ recognition tasks, comprising $5n + 5(n+1)$ positive and $100 + 5n(n-1)$ negative samples. In total, we utilized 3,155 and 2,665 test samples for the single-concept and multi-concept scenarios, respectively.

**Question Answering** All visual tasks share the same data composition. For a scenario with $n$ concepts, each of the 5 single concept test images contributes 5 QA pairs, resulting in $5n$ QA pairs for single concept images. Each of the 5 multi-concept images, assuming they contain all n concepts, generates $2^n - 1$ QA pairs (corresponding to the non-empty subsets of an n-element set). To sum up, for each task, a scenario with n concepts can have up to $5(n + 2^n - 1)$ QA pairs for testing. For text-only QA, an n-concept scenario contains $5n$ single-concept QA pairs and 5 multi-concept QA pairs, resulting in $5n + 5$ text-only QA pairs for testing.

**Positive Examples**

*1 -- Positive recognition*
    Question: <image1> Can you recognize  in this photo ?
    Answer:  Yes,  is in the photo.

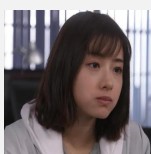

*2 -- Visual question answer*
    Question: <image1> How would you describe  in this photo?
    Answer: is seated in an office setting, her gaze directed off to
          the side, a serious expression on her face.

<image1>

**Negative Examples**

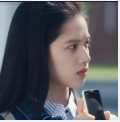

*3 -- Random negative recognition*
    Question: <image2> Can you recognize  in this photo ?
    Answer:  I have not spotted  in this photo.

<image2>

*4 -- Joint negative recognition*
    Question: <image3> Can you recognize  in  this photo ?
    Answer:   is not  in this photo.

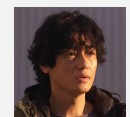

<image3>

Figure 10: **Example of training data for .** <image1> is a photo of , <image2> is randomly selected from the CC12M Changpinyo et al. (2021), and <image3> is selected from the same multi-concept scenario.

**Captioning** In an n-concept scenario, all test images—comprising $5n$ single-concept images and 5 multi-concept images—are utilized for concept experiments. We prompt the model to generate captions for each image and quantitatively assess the model's captioning capability based on the presence of identifiers within the images. This provides a metric for evaluating the model's ability to accurately caption and recognize concepts in both single and multi-concept settings.

### H.3 IMPLEMENTATION DETAILS

We use LLaVA-1.5-13B Liu et al. (2024) as the VLM backbone for all experiments. During training, the $n$ concepts within a single scenario are jointly trained in one pass. For each concept, all 10 images from the training set are used, with a batch size of 1. Training is conducted for 15 epochs, and at the end of the final epoch, we save the token embeddings corresponding to each concept and the parameters of the LM head. Empirically, we set $\tau = 0.32$ and $\gamma = \frac{100}{256 \times 256}$. The training process consists of two phases. First, we initialize the parameters using $k$-means clustering Hartigan & Wong (1979) with Euclidean distance. Then, during the VQA-based training phase, we optimize the model using AdamW Kingma (2014) with a learning rate of 0.001, applying the standard masked language modeling (MLM) loss. Each epoch consists of an average of $250 + 100(n-1)$ steps per concept. For testing, due to the stochastic nature of language model generation and the randomness in test set composition (especially for recognition tasks), each test task is performed three times using the same three fixed seeds for all scenarios. The average result across the three runs is reported. All experiments are conducted on 80GB A100 GPUs.

We find that concept token initialization is crucial as misalignment with the tokenizer's embedding distribution can destabilize training. We normalize the vector norms of concept tokens $K^1, \ldots, K^k$ (denoted $K_*$) from k-means. To align with tokenizer embeddings $K_o$, adjusted tokens are:

$$\hat{K_*} = \frac{K_*}{||K_*||} \cdot K_o \tag{6}$$

### H.4 RECOGNITION QUESTIONS TEMPLATE

During the concept training phase, we follow the positive and negative recognition templates described in Yo'LLaVA Nguyen et al. (2024), using 30 positive and 30 negative templates, respectively. Specifically, for each positive training image, we randomly select 5 QA pairs from the positive recognition templates due to the limited number of positive images. For the joint negative recognition images, we select 10 QA pairs per image from the negative recognition templates. In contrast, for randomly selected images, we only assign one QA pair from the templates. This ensures balanced and diverse training samples while leveraging the templates effectively. Some examples of constructed positive and negative conversations are shown in Fig. 8, 9, 10.

In the test phase, each test uses the query: "Can you see <concept$_i$>in this photo? Answer the question using a single word Yes or No.", where concept$_i$ represents a single concept or several concepts in a multi-concept scenario.

### H.5 CAPTIONING QUESTIONS TEMPLATE

We only use the image captioning question in the test phase. For single-concept captioning and multi-concept captioning, we use the same template, that is, "Can you see <concept$_1$>...<concept$_m$> in the image? Don't answer the question, but remember it, and only respond with a detailed caption for the image. Your caption:".

### H.6 ADDITIONAL QUALITATIVE RESULTS

We provide additional qualitative results for visual question answering, image captioning, and multiple-choice questions, as follows:

1. In Fig. 11, we demonstrate the performance of different models in 3-concept scenarios within the banner Fig. 1.

2. In Tab. 12, 13, we provide examples of VQA in the two-concept scenarios.

3. In Tab. 14, 15, we provide examples of VQA in the three-concept scenarios.

4. In Tab. 16, 17, this VQA example has four concepts.

5. In Tab. 13, we compare the personalized captions of Yo'LLaVA Nguyen et al. (2024) and MC-LLaVA for single-concept scenarios. Both use the captioning questions template provided in Sec. H.5.

6. In Tab. 14, we show the personalized captions of MC-LLaVA in multi-concept scenarios, using the captioning questions template provided in Sec. H.5.

7. In Tab. 18, we show a snapshot of multiple-choice question answering for the lion <A>.

## I LIMITATION AND FUTURE WORK

MC-LLaVA enhances the capacity for personalized interactions in vision-language models, particularly excelling in multi-concept scenarios. However, it is essential to acknowledge several limitations, which can serve as future directions. Firstly, while MC-LLaVA leverages visual information to facilitate the accurate and efficient integration of new concepts into VLMs, the current process still necessitates training, which poses certain challenges for real-world deployment. A promising avenue for future research is to explore the possibility of integrating new concepts to the model without training. Secondly, while our multi-concept dataset pioneers task-level evaluation in VLM personalization, the field lacks comprehensive benchmarks that encompass a larger scale and capability dimensions. This limitation restricts our assessment of capability-level VLM personalization. Future work could define the capabilities that models should be evaluated on and propose more comprehensive benchmarks that assess various aspects of personalization capabilities in VLMs. Finally, as illustrated in Figure 17, scenarios composed of visually near-identical concepts (e.g., identical twins) remain challenging: even humans, MC-LLaVA, and strong proprietary models such as Gemini 3 and GPT-4o often fail to reliably distinguish them, leading to concept confusion. Addressing such inherently ambiguous cases remains an important open challenge for future work.

| | | | |
|---|---|---|---|
| <Liam> <Noah> <Emma> | Question: What are <Liam> and <Noah> wearing in this image? | MC-LLaVA: <Liam> is wearing a green-and-white checkered shirt. <Noah> is wearing a blue and black striped long-sleeve shirt. | Yo'LLaVA: <Liam> is wearing a green and white plaid shirt, ~~while <Liam> is~~ sporting a blue and black striped long-sleeve top. | LLaVA: In the image, ~~there are two individuals~~ who appear to be children. The child on the left is wearing a green and white shirt, ~~and the child on the right is wearing a blue jacket and a scarf.~~ |

Figure 11: The performance of the scenario displayed in the banner across different models.

**GPT-4o Caption Evaluation Template**

You will act as an evaluator that scores a caption by comparing it to a ground truth caption. You will receive two pieces of information:
Ground Truth: The reference caption containing special identifiers (e.g., <A>, , etc.)     Generated Caption: The caption that needs to be evaluated

Your evaluation will follow these strict rules:
1. Identifier Check (Critical):
If the generated caption is missing any identifiers from the ground truth, score proportionally to the percentage present
If identifiers are present but their relationships are incorrect/mixed up, score 0
Only proceed to other criteria if identifiers are present and correctly related
2. If identifier check passes, evaluate on three key criteria:
Accuracy (40%): How factually correct is the caption compared to ground truth?
Relevance (30%): How well does the content align with the ground truth's main points?
Helpfulness (30%): How effectively does it convey the key information?
3. Score each criterion from 0-10, where:
0: Completely incorrect/irrelevant/unhelpful
10: Perfect match with ground truth
Final score will be weighted average of the three criteria (only if identifier check passes).

Please provide:
Identifier Check Result (Pass/Fail/Partial)
Individual criteria scores (if applicable)
Overall Score (0-10)
Brief explanation of scoring rationale

Example:
Ground Truth: "<A> is wearing a red hat and  has blue shoes"     Generated Caption : "<A> wears a green hat and  has blue shoes"
Output:
Identifier Check: Pass
Accuracy: 5/10 (hat color wrong)     Relevance: 9/10 (describes same elements)     Helpfulness: 8/10 (conveys main info)
Overall Score: 7/10
Explanation: All identifiers present and correct. Minor accuracy issue with hat color, but maintains good relevance and helpfulness.

Ground Truth: {{your ground truth}}   Generated Caption: {{your caption}}

Figure 12: The caption evaluation template prompted to GPT-4o

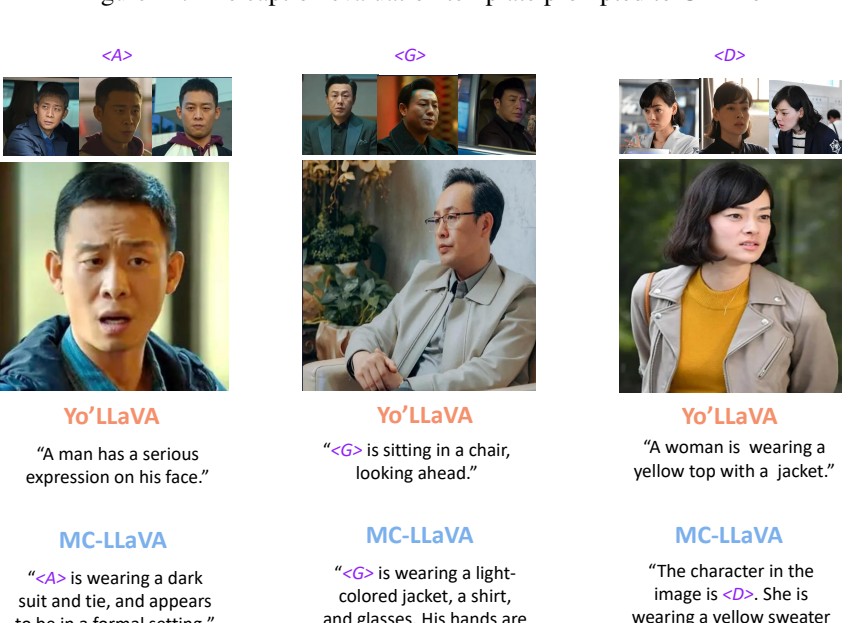

Figure 13: Comparison of personalized captioning between Yo'LLaVA and MC-LLaVA.

**MC-LLaVA**

<D>: 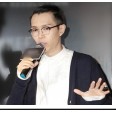    <K>: 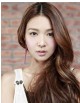

▷ *Visual Conversation* — (Note: Single-concept)

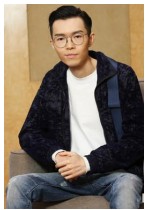    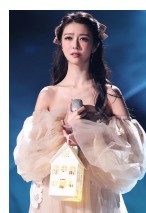

| Question: | What is <D> wearing in the image? | Is <K> in this photo? If yes, describe her location and the scenario. |
|---|---|---|
| MC-LLaVA: | <D> is wearing a dark blue velvet jacket, a white t-shirt, and light blue denim jeans. | Yes, I can confirm that <K> is indeed in the photo. <K> is standing against a blue background with a soft, diffused light that creates a gentle glow around her. |

▷ *Visual Conversation* — (Note: Multi-concept)

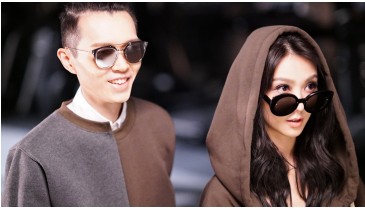    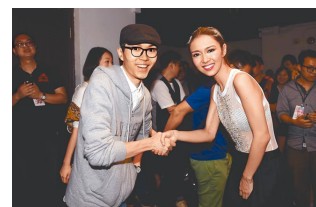

| Question: | What are <D> and <K> wearing in the image? | Can you see <D> and <K> in this picture? If so, please describe their attire and the scenario. |
|---|---|---|
| MC-LLaVA: | <D> is wearing a brown and gray sweater, while <K> is wearing a brown and gray hooded garment. And they both wear sunglasses. | Yes, I can see them here. <D> is wearing a gray cap and gray pants. <K> is wearing a black and white plaid shirt. <D> and <K> are shaking hands in the image. |

Table 12: Examples of MC-LLaVA's abilities with learned concept <D> and <K>.

**MC-LLaVA**

<J>: 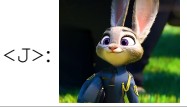 <N>: 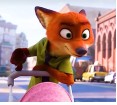

▷ *Visual Conversation* — (Note: Single-concept)

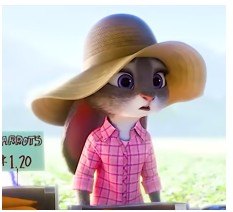 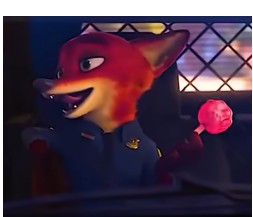

| Question: | What is <J> wearing in the image? | What is <N> holding in his hand? |
|---|---|---|
| MC-LLaVA: | <J> is wearing a pink and white checkered shirt and a straw hat. | <N> is holding a pink lollipop in his hand. |

▷ *Visual Conversation* — (Note: Multi-concept)

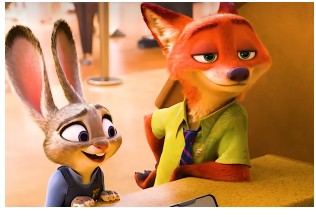 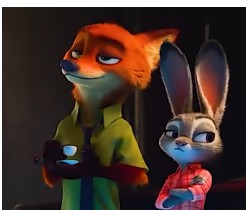

| Question: | What are <J> and <N> doing in the image? | What are <J> and <N> doing in the image? |
|---|---|---|
| MC-LLaVA: | <J> and <N> are sitting at a table, looking at a tablet. <N> is wearing a green shirt with a tie, and <J> is wearing a blue shirt. They both appear to be engaged in whatever is displayed on the tablet. | <N> and <J> are standing next to each other in the image. <N> is wearing a green shirt and a blue tie, and he has a cup in his hand. <J> is wearing a red and white checkered shirt. They both appear to be looking off to the side with serious expressions. |

Table 13: Examples of MC-LLaVA's abilities with learned concept <J> and <N>.

**MC-LLaVA**

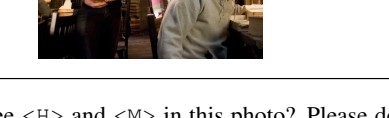

▷ *Visual Conversation* — (Note: Single-concept)

| Question: | What is <M> doing? | Is <R> in this photo? If yes, describe his location and the scenario. |
|---|---|---|
| MC-LLaVA: | <M> is holding a green glass flask in her left hand and a clear glass flask in her right hand, suggesting she might be conducting an experiment or a study. | Yes, <R> is in this photo. <R> is standing in a classroom, his face contorted in an expression of shock. His eyes are wide and his mouth is slightly open, suggesting he has just been startled or caught off guard. |

▷ *Visual Conversation* — (Note: Multi-concept)

| Question: | What are <H>, <M>, and <R> doing in this photo? | Can you see <H> and <M> in this photo? Please describe the current scenario. |
|---|---|---|
| MC-LLaVA: | <H> and <M> are peeking around a corner, with <M> wearing a pink hoodie. <R> is on the far right, holding a small animal in his hand. | Yes, I can confirm that <H> and <M> are indeed in the photo. <H> is sitting at a table with books, while <M> is standing nearby in a library setting. <R> seems to be in the picture too, holding a golden egg in his hand. |

Table 14: Examples of MC-LLaVA's abilities with a learned concept <H>, <M> and <R>.

**MC-LLaVA**

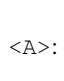  <A>:  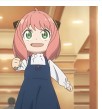  <L>:  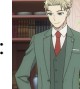  <Y>:  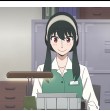

▷ *Visual Conversation* — (Note: Single-concept)

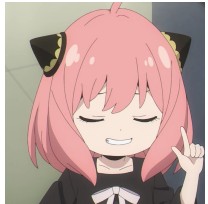  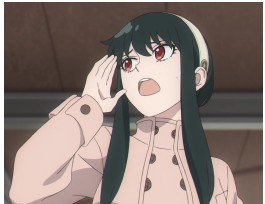

| Question: | What is <A> wearing on her head? | What is <Y> doing with her hand in the image? |
|---|---|---|
| MC-LLaVA: | <A> is wearing black and gold horn-like headpieces. She has pink hair. | <Y> is holding her hand up to her face, possibly in surprise or to block her face. She is wearing a pink jacket with black details and has long black hair. |

▷ *Visual Conversation* — (Note: Multi-concept)

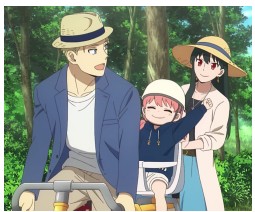  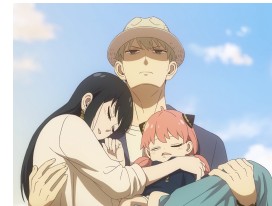

| Question: | What activity are <A>, <L>, and <Y> engaged in? | What are <A>, <L>, and <Y> doing in the image? |
|---|---|---|
| MC-LLaVA: | <L> is riding a bicycle with <A> in a stroller attached to it and <Y> standing behind them. He is wearing a hat and a blue jacket. | <A>, <L>, and <Y> are in a family pose. <A> is sleeping on <L>'s shoulder, and <Y> is holding <A>. They are under a clear sky. |

Table 15: Examples of MC-LLaVA's abilities with a learned concept <A>, <L> and <Y>.

**MC-LLaVA**

<Y>:   <Z>:   <M>:   <Q>:

▷ *Visual Conversation* — (Note: Single-concept)

| Question: | What is <Z> wearing in the photo? What is he doing? | Can you describe the scenario in the photo? |
|---|---|---|
| MC-LLaVA: | <Z> is wearing a green jacket over a white shirt with horizontal stripes. He has a neutral expression on his face and is making a hand gesture with his right hand. | In the photo, we see <M> sitting at a kitchen counter, her hand resting on her chin in a thoughtful pose. The overall scenario suggests a moment of contemplation or preparation. |

▷ *Visual Conversation* — (Note: Multi-concept)

| Question: | What activity are <Y>, <Z>, <M> and <Q> engaged in? | Can you provide a detailed description of what <Y>, <Z>, <M> and <Q> are wearing in this photo? |
|---|---|---|
| MC-LLaVA: | <Y>, <Z>, <M> and <Q> are sitting together on a red couch, taking a selfie with a phone. | Yes, <Y> is wearing a red dress with a belt, <Z> is wearing a black and white checkered shirt with blue jeans, <M> is wearing a light pink dress, and <Q> is wearing a blue checkered suit. |

Table 16: Examples of MC-LLaVA's abilities with learned concept <Y>, <Z>, <M> and <Q>.

**MC-LLaVA**

<H>:    <P>:    <Z>:    :

▷ *Visual Conversation* — (Note: Single-concept)

| Question: | What is <Z> doing in the image? | What is  wearing on her head? |
|---|---|---|
| MC-LLaVA: | <Z> is standing in front of a mailbox, looking displeased. |  is wearing a clear helmet with a pink flower on top. |

▷ *Visual Conversation* — (Note: Multi-concept)

| Question: | What festive accessory are <H>, <P>, <Z> and  wearing? | What are <H>, <P>, <Z> and  doing in the image? |
|---|---|---|
| MC-LLaVA: | <H>, <P> and <Z> are wearing red Santa hats on their heads, but  wears a helmet with a flower on it. | <H>, <P> and  are standing together with happy expressions, but <Z> looks bored. |

Table 17: Examples of MC-LLaVA's abilities with learned concept <H>, <P>, <Z> and .

**MC-LLaVA**

"<R>, <Z>, and <P> are sitting outdoors with a laptop. <R> is wearing a green jacket and jeans. <Z> is wearing …"

**MC-LLaVA**

"<A>, <G>, and <M> are standing on a beach with the ocean in the background. They looks all surprised."

**MC-LLaVA**

"In the image, I can see <L> , <W> , and <N> . <L> is shading her eyes with her hand, <W> is wearing a red and white outfit with a bow, and <N> is in a pink and white dress with a bow. They are standing in a grassy field with flowers."

Figure 14: Personalized caption of multi-concept with MC-LLaVA.

**Text-only Question-Answering** *(No image is given as input)*
*[due to limited space, only a fraction of the questions are shown here]*

**Question 1:** Is <A> a lion or a cat?
    A. A lion
    B. A cat
    *Correct Answer: A*
**Question 2:** What is <A>'s mane color?
    A. brown
    B. black
    *Correct Answer: A*
**Question 3:** What color are <A>'s eyes?
    A. green
    B. blue
    *Correct Answer: B*
**Question 4:** Is <A>'s demeanor lively or serious?
    A. lively
    B. serious
    *Correct Answer: A*
**Question 5:** Does <A> have a large or small mane?
    A. large
    B. small
    *Correct Answer: A*

**Visual Question-Answering**
*[due to limited space, only a fraction of the questions are shown here]*

**Question 1**: What is <A> looking at in the image?
    A. His claws
    B. The sky
    *Correct Answer: A*

**Question 2**: What is <A> doing in the image?
    A. Sleeping
    B. Dancing
    *Correct Answer: B*

Table 18: Example of multiple choice question answering.

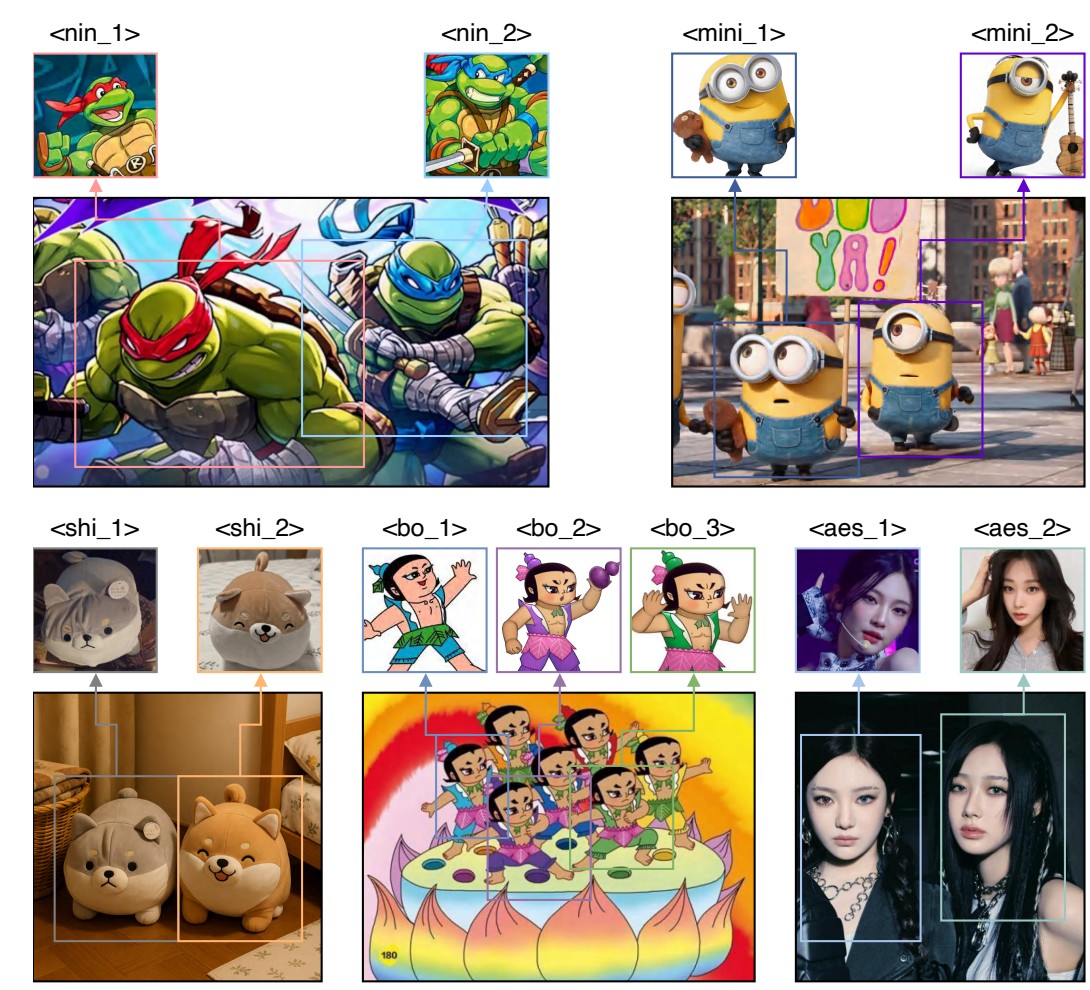

Figure 15: Five newly collected multi-concept scenarios, where the concepts within each scene exhibit strong visual similarity.

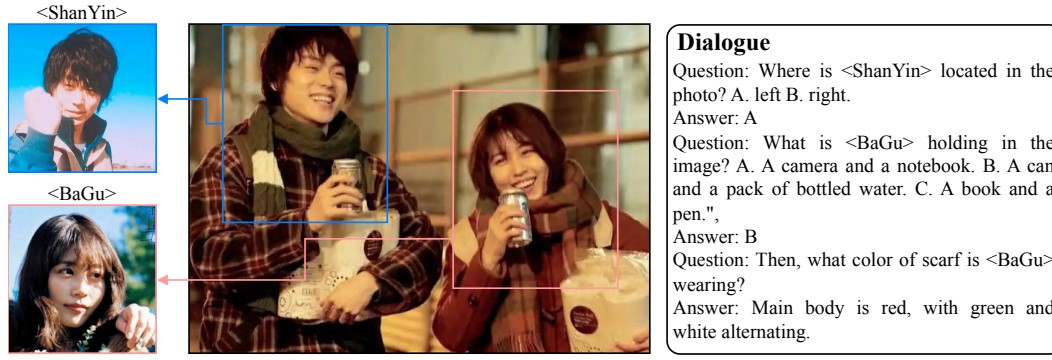

**Dialogue**

Question: Where is <ShanYin> located in the photo? A. left B. right.
Answer: A
Question: What is <BaGu> holding in the image? A. A camera and a notebook. B. A can and a pack of bottled water. C. A book and a pen.",
Answer: B
Question: Then, what color of scarf is <BaGu> wearing?
Answer: Main body is red, with green and white alternating.

Figure 16: A sample from the multi-turn dialogue evaluation data.

## J    NOTES ON THE NEW DATASET

As shown in Figure 15, we further curate five multi-concept scenarios, each containing two or three concepts with similar visual attributes. These scenarios form a small toy dataset used to evaluate how well different models can distinguish between and remember visually similar concepts.

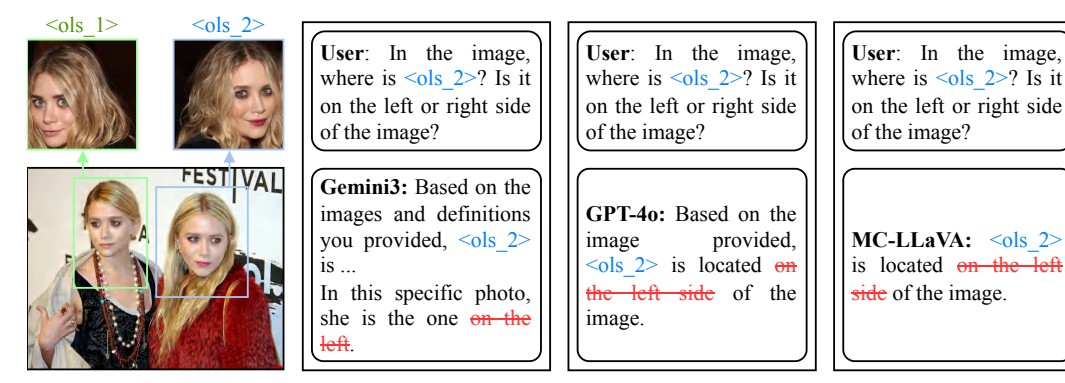

Figure 17: Showcase of the performance of existing models in a scenario with concepts of extremely similar visuals.

As illustrated in Figure 16, we constructed a specialized test set involving 5 scenarios with GPT-generated dialogues (2-3 turns each). The questions include both visual multiple-choice and open-ended QA, and we adopt an LLM-as-a-judge protocol to aggregate model scores.

