# OpenReview forum: "MC-LLaVA: Multi-Concept Personalized Vision-Language Model"
_ICLR.cc/2026/Conference — Submitted to ICLR 2026_

### Official Review · Reviewer_BhVC · 2025-10-30

**Soundness:** 2
**Presentation:** 2
**Contribution:** 2
**Rating:** 4
**Confidence:** 5

**Summary:**

This paper proposes MC-LLaVA, a method for multi-concept Large Vision-Language Model personalization. The contribution of this paper includes:
1. A dataset for the task of multi-concept LVLM personalization.
2. A method to address the problem of multi-concept LVLM personalization, including joint training to eliminate the high-quality negative samples, and visual prompting to enable LVLM personalization for multiple concepts.

The dataset contribution is significant to the field of Multi-concept LVLM personalization, since this field lacks a large dataset for training and evaluation, as the current datasets are quite simple and mostly are focused on single-concept personalization.

**Strengths:**

1. The dataset contribution of this paper is crucial. There are not many available datasets for LVLM personalization, especially in terms of the multi-concept setting. Most of the dataset from YoLLaVA and MyVLM focuses on a single concept of LVLM personalization, and most images from that dataset only contain a single identity.
2. The experiments indicate the superior performance of the proposed methods compared with the Baselines, including YoLLaVA, MyVLM, and RAP-MLLM.
3. The joint training mechanism can reduce the need for high-quality negative samples, which is essential, since the YoLLaVA paper requires a huge dataset (Laion) to retrieve the high-quality negative samples.

**Weaknesses:**

There are several weaknesses in this paper:

1. Lack of novelty: The whole framework is based solely on YoLLaVA for personalization, including the system prompt design and test-time training for each personalized concept.
2. The mark indicator is the one that enables multi-concept recognition; however, it is not new in terms of LVLM personalization. The Personalization Toolkits [1] paper has already used this idea to mark the personalized concept. The paper should discuss this.
3. Inefficient method: This method is inefficient; it requires both test-time fine-tuning and a memory bank to store the personalized CLIP features for multi-concept personalization, especially in the case of the continual setting, while new concepts arrive at test time. To me, it looks like we are both using test-time fine-tuning and also using a RAG mechanism to address the LVLM personalization. The test-time fine-tuning takes a lot of time, showing this method is less efficient compared to the RAG-based method, such as Personalization Toolkits [1] and RAP-MLLM [2]. Also, this paper lacks a comparison with the LVLM models [3], which accept multiple images as input, and can accept multiple images of the same concept as inputs.
4. Dataset: How can we ensure the quality of the dataset for the QA task? If there are multiple IDs in the same image, is there any chance that the LVLM can have a correct answer on the QA task, but it will be wrong in the recognition task with the same concept? From what I see in the dataset construction, we are only counting the correct answer, similar to the YoLLaVA paper.
And in addition, it is hard to describe the location of a concept in the photo for the grounding task, especially in the case of multiple objects in the image. In computer vision, the position of an object should be indicated by either a segmentation mask or a bounding box.
It is better for the grounding task to be compared with grounded-capability LVLM [4].

Reference:

[1] Seifi, Soroush, et al. "Personalization Toolkit: Training Free Personalization of Large Vision Language Models." arXiv preprint arXiv:2502.02452 (2025).

[2] Hao, Haoran, et al. "RAP: Retrieval-Augmented Personalization for Multimodal Large Language Models." Proceedings of the Computer Vision and Pattern Recognition Conference. 2025.

[3] https://huggingface.co/docs/transformers/en/model_doc/llava_next

[4] Zhang, Tao, et al. "Omg-llava: Bridging image-level, object-level, pixel-level reasoning and understanding." Advances in neural information processing systems 37 (2024): 71737-71767.

**Questions:**

Based on the weaknesses, there are several follow-up questions:
1. What is the fine-tuning time of MC-LLaVA? How do we choose the concept to be the negative samples when we want to train new concepts?
2. What are the results compared with the Personalized Toolkit [1] and some LVLMs that accept multiple images as input [2]?
3. If there are multiple IDs in the same image, is there any chance that the LVLM can have a correct answer on the QA task, but it will be wrong in the recognition task with the same concept?
4. What is the difference between the S and M versions of YoLLaVA baselines?
5. Is the comparison with GPT-4o P fair? Because the GPT-4o only receives the text prompt as the knowledge of the concept, and MC-LLaVA trains on the visual appearance of the concept, thus making the comparison of MC-LLaVA with the LVLM model that receives multiple images as input less efficient.
6. What is the performance of MC-LLaVA on multi-turn conversation?

[1] Seifi, Soroush, et al. "Personalization Toolkit: Training Free Personalization of Large Vision Language Models." arXiv preprint arXiv:2502.02452 (2025).

[2] https://huggingface.co/docs/transformers/en/model_doc/llava_next

---

> ### Author Response · Authors · 2025-11-21
> **Rebuttal by Authors (Part 1)**
>
> We sincerely thank the reviewer for the detailed review and for recognizing the **crucial nature of our dataset contribution** to the field. We also appreciate your acknowledgment that our method outperforms baselines like Yo'LLaVA and RAP-MLLM while eliminating the need for massive negative sample retrieval. We address your concerns below.
>
> **W1**: Technical novelty.
>
> We respectfully but disagree with the assessment that our framework is "based solely on Yo'LLaVA." This perspective overlooks the existing issue of Yo'LLaVA in a multi-concept scenarios (as shown in Fig. 1, Tab. 2-3), as well as specific design we propose to address these issues. Beyond the points you mentioned, we also present the following novel designs:
> - **Inter-Concept Negative Sampling**: A novel joint training objective that pairs images of Concept A with Concept B within the same batch. This compels the model to learn discriminative features and reduces the need of hard negative examples, demonstrating enhanced robustness in visually similar concept scenarios.
> - **Visual Token Initialization**: We leverage visual features (obtained via k-means) to initialize textual embeddings. This architectural choice bridges the visual-text gap, significantly accelerating convergence (as shown in Fig. 5) and reducing data dependency.
> - **Personalized Visual Prompt**: This complements the personalized textual prompt, constructing a complete dual joint training framework, which further augments the model's performance across various tasks (see Tab. 4).
>
> **W2 & Q2**: Discussion of related work.
>
> We respectfully point out that Reference [1] was published **only on arXiv** and its **code is not open-sourced**.
>
> According to **ICLR 2026 Policy**:
> > "Note that arXiv is not considered a peer-reviewed venue. As such, authors are not required to compare to papers solely on arXiv.... the lack of such comparisons cannot be a basis for rejection."
>
> Therefore, we are not obliged to compare with it. However, to further alleviate your concerns, we emphasize the fundamental differences in our approach. [1] solely relies on extra segmentation models to extract visual features, creating a rigid pipeline where the VLM passively receives retrieved features. In contrast, MC-LLaVA employs a dual framework. Our Personalized Visual Prompt is not an isolated retrieval result; it is an **explicit injection** of concept feature that works synergistically with the **implicit visual features** initialized within the Personalized Textual Prompt. This mechanism allows explicit spatial guidance and implicit semantic alignment to optimize jointly during fine-tuning, whereas [1] is limited by the frozen capabilities of the external segmenter and the VLM. Practically, we compare MC-LLaVA and [1] in two settings:
>
> - **Visually similar scenarios**: As detailed in Tab. 1 of the global response, to cope with core the challenge of disambiguation in multi-concept scenarios, our fine-tuned embeddings provide significantly finer-grained discrimination than the training-free retrieval features in [1].
> - **General effectiveness**: We notice that inference prompt used in [1] and MC-LLaVA is different. To achieve fair evaluation, we adopt their prompt on our method because the code is not open-sourced. As detailed in Tab. 2 of the global response,  maintaining the same length of input token length, MC-LLaVA consistently outperforms [1] across datasets, validating that our parametric integration is more robust than context-window injection.

---

> ### Author Response · Authors · 2025-11-21
> **Rebuttal by Authors (Part 2)**
>
> **W3 & Q1**: Method efficiency and scalability.
>
> **Efficiency among FT Methods**: We would like to point out that MC-LLaVA is highly efficient within the category of finetune-based methods. Our "Visual Token Initialization" strategy significantly accelerates convergence and reduces the need for hard negative samples. As shown in the table below, the training time is comparable to Yo'LLaVA, while we achieve multi-concept capability which Yo'LLaVA lacks.
>
> | Methods   | Training Time/Concept |
> |-----------|-----------------------|
> | Yo'LLaVA  | ~15 min               |
> | MC-LLaVA  | ~16 min               |
>
> **FT vs. RAG**: While RAG-based methods (e.g., RAP-MLLM) have faster enrollment, they suffer in performance. As evident in Table 3, MC-LLaVA outperforms RAP-MLLM in recognition (+0.16), visual localization (+0.11), and VQA (+0.25 BLEU). A comparative analysis of MC-LLaVA and [1] can be found in Tab. 1 and Tab. 2 of the global response. MC-LLaVA achieves better performance across bench and even in the newly added visual similar concept scenarios. We prioritize accuracy; a fast model that confuses concepts is not useful.
>
> **Model Agnostic**: To further demonstrate robustness, we applied our method to LLaVA-OneVision (LLaVA-OV)[5], a stronger backbone that accepts multiple images. As detailed in Tab. 4 of the global response, MC-LLaVA consistently enhances personalization in this advanced architecture, proving our paradigm is model-agnostic.
>
> **Negative sample strategy**: Regarding the training of new concepts, we selected 5 representative scenarios from the MC-LLaVA Benchmark. For each incoming concept, we expand the vocabulary and optimize only the newly added tokens. We simply select other concepts within the same scenario group that serve as natural negative samples for the current target. Evaluation after sequential training confirms that this simple strategy yields enough robust performance for each concept.
>
> | Methods                          | MC-LLaVA |MC-LLaVA|MC-LLaVA|MC-LLaVA|
> |----------------------------------|----------|--------|--------|-------|
> |                                  | Rec.     | VQA Acc | QA Acc | VG    |
> | MC-LLaVA (Training separately)   | **0.882**| **0.891** | **0.700** | **0.716** |
> | MC-LLaVA (Training continually)  | 0.833    | 0.884  | 0.633  | 0.710 |
> | LLaVA + Prompt                   | 0.577    | 0.813  | 0.548  | 0.525 |
>
> **W4 & Q3**: Assurance of dataset quality.
>
> **For concern about QA correctness vs. Recognition failure**:
>
> In our dataset construction process, we manually checked the generated answer to ensure quality. Furthermore, we follow the prompt design in Yo'LLaVA and expand it to multi-concept scenarios, which could not be answered correctly without actually understanding the concept. Empirically, the mismatch rate is extremely low (< 1%), meaning correct QA answers are strongly correlated with correct recognition.
>
> **For the grounding task**:
>
> We clarify that our proposed "Personalized Visual Grounding" serves as a proxy metric to evaluate the degree of concept personalization, not pursuing fine-grained pixel level segmentation (as in [6]). correctly binding a specific identity to a spatial region (e.g., "The \<dog\> is on the left") sufficiently verifies that the model effectively "sees" and disambiguates the concept from others.
>
> To further explore MC-LLaVA's performances, we adopt the OMG-LLaVA as backbone following the same experimental settings. Experiments show that MC-LLaVA's Visual Grounding performance improves further. This confirms that our framework naturally inherits and leverages the advancing grounding capabilities of base models.
>
> | Methods                  | MC-LLaVA: VG |
> |--------------------------|----------------------|
> | LLaVA+P                  | 0.528                |
> | OMG-LLaVA+P              | 0.533                |
> | MC-LLaVA (LLaVA)         | 0.723                |
> | MC-LLaVA (OMG-LLaVA)     | **0.754**            |

---

> ### Author Response · Authors · 2025-11-21
> **Rebuttal by Authors (Part 3)**
>
> **Q4**: Clarification of Yo'LLaVA experimental setting.
>
> - **YoLLaVA-S (Single)**: Loads parameters for a single concept and is inherently incapable of handling multi-concept queries.
> - **YoLLaVA-M (Multi)**: Simply loads parameters for multiple separately trained concepts and expands the classification head. While it supports multi-concept scenarios, it suffers from severe performance degradation due to optimization conflicts (see Tab. 3). This failure prompted us to adopt a joint training approach.
>
> **Q5**: Discussion of comparison fairness.
>
> We conducted an additional experiment using GPT-4o + Image Prompts (IP). While providing reference images improves GPT-4o's performance, it comes at a skyrocketing token cost (~1k tokens per query due to multi-image processing). As shown in the table below, even with Image Prompts, MC-LLaVA remains competitive and more efficient during inference.
>
> | Methods        | Tokens | MC-LLaVA |MC-LLaVA|MC-LLaVA| Yo'LLaVA | MyVLM      |
> |-----------------------------|---------|----------|---------|--------|----------|------------|
> |                             |         | Rec.     | VQA Acc | QA Acc | Rec.     | Captioning |
> | MC-LLaVA                    | 10^1    | **0.878**| 0.890   | 0.709  | **0.947**| **0.975**  |
> | GPT-4o + text prompt        | 10^1    | 0.781    | 0.889   | 0.702  | 0.856    | 0.969      |
> | GPT-4o + image prompt       | 10^3    | 0.849    | **0.920** | **0.801** | 0.909    | **0.975**  |
>
> **Q6**: Analysis MC-LLaVA performances on multi-turn conversation.
> Thank you for this insightful question. Since existing benchmarks [1,2,3,4] lack multi-turn evaluation for personalization, we constructed a specialized test set involving 5 scenarios with GPT-generated dialogues (2-3 turns each). As shown below, MC-LLaVA demonstrates superior stability in multi-turn settings compared to LLaVA+P, proving that our Visual Prompt serves as a stable context anchor.
>
> | Methods   | Multi-turn VQA Acc | Single-turn VQA Acc |
> |-----------|--------------------|---------------------|
> | LLaVA + P | 0.771              | 0.801               |
> | MC-LLaVA  | **0.853**          | **0.875**           |
>
> [1] Personalization Toolkit: Training Free Personalization of Large Vision Language Models, Arxiv 2025.
>
> [2] MyVLM: Personalizing VLMs for User-Specific Queries, ECCV 2024.
>
> [3] Yo'LLaVA: Your Personalized Language and Vision Assistant, NIPS 2024.
>
> [4] RAP: Retrieval-Augmented Personalization for Multimodal Large Language Models, CVPR 2025.
>
> [5] LLaVA-OneVision: Easy Visual Task Transfer, TMLR 2025.
>
> [6] Omg-LLaVA: Bridging image-level, object-level, pixel-level reasoning and understanding, NIPS 2024.

---

> > ### Comment · Reviewer_BhVC · 2025-11-24
> >
> > Thank you to the authors for the comprehensive explanation. However, after reviewing the rebuttal, I still have the following concerns:
> > 1. **Grounding task**: I disagree with the claim that textual location descriptions are sufficient to specify the position of a concept. While such descriptions (e.g., “on the left”) may work in simple images containing only two or three objects, this paper targets Multi-Object Personalization, where images may contain five or six objects. In these more complex scenarios, explicit grounding signals, such as segmentation masks or bounding boxes, can provide substantially greater interpretability and clarity than purely textual descriptions.
> >
> > 2. **Comparison with GPT-4o**: Why are the results of MC-LLaVA way worse than GPT-4o in terms of VQA, while being much better in terms of the recognition metric in the reported table?

---

> > > ### Author Response · Authors · 2025-11-25
> > > **Follow-up for Reviewer BhVC**
> > >
> > > Dear Reviewer BhVC,
> > > Thanks again for your valuable and constructive review, which has inspired us to improve our paper further substantially.
> > >
> > > 1. **Grounding task**:
> > >
> > > We appreciate this insightful comment; however, we clarify our scope and design rationale:
> > >
> > > - **Current Scope**: Our work currently focuses on scenarios with **2 to 4 concepts**, which covers the vast majority of typical user-assistant interactions (e.g., family photos, movie scenes). In this specific regime, our experimental results (**MC-LLaVA: 0.723 v.s. LLaVA: 0.458**) show that relative positional descriptions are sufficient to evaluate concept differentiation.
> > > - **Future Work**: We acknowledge that scaling to highly crowded scenes (5+ objects) requires finer-grained output formats, as the concept data is hard to curate and annotate. Integrating pixel-level segmentation output to support dense-object personalization is a promising direction we plan to explore in future work.
> > >
> > > 2. **Comparison with GPT-4o**:
> > >
> > > - **Task Format Sensitivity**: Tasks like **Recognition** and **Multiple-Choice (Choice-V/T)** have relatively constrained output formats, which often mask the reasoning gap between models. In these specific personalization metrics, MC-LLaVA effectively injects user concepts, allowing it to outperform even GPT-4o (Rec: **0.878** vs. 0.849). To strictly evaluate the model's upper limits, we specifically utilized **Free-form VQA**, as its open-ended nature better reveals the disparities between open-source and closed-source architectures.
> > > - **Backbone Disparity**: GPT-4o is a **state-of-the-art closed-source** model with vastly superior general knowledge and reasoning capabilities compared to the LLaVA backbone used in our main experiments. Free-form VQA relies **heavily** on these inherent general capabilities. Therefore, GPT-4o's lead in VQA reflects its stronger base model foundation rather than a deficiency in our personalization method.
> > > - **Inherited Boundaries vs. Core Contribution**: We respectfully argue that the performance ceiling imposed by the LLaVA-1.5 backbone **should not overshadow our core contributions** (i.e., the Dual Multi-Concept Framework). As shown in Tab. 4 of the Global Response, when we switch to a stronger backbone (**LLaVA-OV**), the VQA performance significantly improves (**+4.3%**), demonstrating that our method scales with better base models.
> > >
> > > Once again, we sincerely appreciate your professional review. We hope that this follow-up has fully addressed any remaining concerns, and we look forward to your continued support.

---

> ### Author Response · Authors · 2025-11-24
>
> Dear reviewer BhVC, As the discussion deadline is approaching, we kindly invite you to share your thoughts and engage in discussion if possible. We truly appreciate the time and effort you've put into reviewing our submission. If there are any points you'd like us to clarify or discuss further, we are more than happy to assist. Thank you again for your valuable feedback and contributions. Best regards, Authors of submission 7804

---

### Official Review · Reviewer_r43i · 2025-11-01

**Soundness:** 4
**Presentation:** 4
**Contribution:** 2
**Rating:** 4
**Confidence:** 3

**Summary:**

This paper investigates multi-concept personalization for Vision-Language Models (VLMs), advancing beyond existing single-concept approaches. To achieve this goal, the authors propose a multi-concept instruction tuning strategy that integrates: Personalized textual prompts (initialized with visual tokens for faster training convergence) and Personalized visual prompts (encoding spatial location information). Additionally, the authors introduce a new multi-concept instruction tuning dataset. Experiments are conducted on their proposed dataset and two benchmarks (Yo’LLaVA and MyVLM), evaluating four tasks: recognition, visual grounding, VQA, and captioning.

**Strengths:**

1. The writing is concise and the method is sound.
2. Significant performance improvements are demonstrated across all three datasets for both single- and multi-concept personalization.

**Weaknesses:**

1. Primary Concern: The core contribution of the paper lies in extending single-concept personalization to multi-concept personalization. The proposed method is technically sound, as it guides each textual prompt with its corresponding visual concept. However, the paper primarily addresses the increasing number of concepts while lacking depth in tackling critical challenges inherent to multi-concept personalization, such as concept disambiguation—for instance, how to differentiate between visually similar objects (e.g., distinguishing between two nearly identical objects).

2. Missing important ablation studies on hyperparameters: 1) K-means clustering: Why is k-means used to generate personalized textual prompts? How is the optimal k (number of clusters) determined? Is k fixed across all concepts, or does it adapt to the variance in visual appearance? 2) Number of images per concept (e.g., minimum required for robust personalization).

**Questions:**

1. Must all textual concept prompts be provided at inference? If so, how does this scale with increasing concepts?
2. How is location information integrated into the VLM?
3. Clarify the role of "Mark number 1" (e.g., is it a placeholder for a mask image or a learned token?).

---

> ### Author Response · Authors · 2025-11-21
> **Rebuttal by Authors (Part 1)**
>
> We sincerely thank the reviewer for the **"Excellent"** rating on soundness and presentation, and for acknowledging the **significant performance improvements** of our method. We appreciate your insightful comments on concept disambiguation and hyperparameters. We address your concerns below.
>
> **W1**: Performances on visually similar concepts.
>
> We respectfully clarify that the MC-LLaVA framework is designed to address the core challenge of disambiguation in multi-concept scenarios:
> - **Discriminative Training**:  Our **Inter-concept Negative Sampling** strategy forces the model to distinguish between co-occurring concepts by pairing Concept A images with Concept B as hard negatives. This moves beyond simple recognition to discrimination, boosting model's robustness handling disambiguation scenarios.
> - **Dual Guidance**: Besides the explicit spatial guidance from **Personalized Visual Prompts**, the **implicit visual features** injected into textual prompts allow the VLM to attend to fine-grained differences during auto-regressive generation.
> - **Quantitative Verification**: Following your suggestion, we constructed a "Hard Case" dataset (5 scenarios with visually similar objects, shown in Fig. 15). As shown in Table 1 of the global response, MC-LLaVA significantly outperforming RAP-MLLM by 0.19 in recognition task. The newly added Personalization Toolkit [1] performs poorly here due to its only reliance on generic visual similarity without discriminative training.
> - **Limitation**: We acknowledge that distinguishing identical objects (e.g., identical twins without extra information) remains an open challenge, which is also hard for humans and the most advanced models. We discuss limitations and showcase such difficult cases in Fig. , reserving it as future work.
>
> **W2**: Analysis of hyperparameters.
>
> - **Why using k-means clustering**: Theoretically, raw visual tokens contain redundant background noise. K-means aggregates them into compact centers, extracting the semantic "essence" while filtering noise. Empirically, as shown in Tab. 9 (Appendix E.3), our ablation study confirms that K-means initialization consistently outperforms Random, Pooling, and PCA methods.
> - **The selection of number of K**:  As shown in Table 2 of the global response,  we provide performances of MC-LLaVA on different value of K. Ablation studies can successfully validate the effectiveness of the design of K-means. Performance improves with k and peaks around k =16. While adaptive k value selection is an interesting direction, empirical results show that a fixed k = 16 providing robust performance across diverse concepts. Following Occam's razor, we choose k = 16 among different datasets and concepts.
> - **Number of images per concept**: We conducted an ablation on training set size (see table below). The performance decline when using eight images is minimal, indicating that our method exhibits high data efficiency. When only having five input images, the performance decreases due to the insufficient volume of data while still surpassing MyVLM on MC-LLaVA benchmark.
>
> | Training images | MC-LLaVA |MC-LLaVA|MC-LLaVA| Yo'LLaVA |
> |-----------------|----------|-------|-------|----------|
> |                 | Rec.     | VQA Acc   | VG    | Rec.     |
> | 5               | 0.811    | 0.839 | 0.677 | 0.893    |
> | 8               | 0.865    | 0.887 | 0.715 | 0.939    |
> | 10              | **0.878**| **0.890** | **0.723** | **0.947** |
>
> [1] Personalization Toolkit: Training Free Personalization of Large Vision Language Models, Arxiv 2025.

---

> ### Author Response · Authors · 2025-11-21
> **Rebuttal by Authors (Part 2)**
>
> **Q1**: Inference settings and scalability.
> - **Inference settings**: Textual concept prompts for all personalized concepts are required. Without their discriminative feature, MC-LLaVA may suffer in visually similar scenarios.
> - **Scalability**: However, scalability is not a bottleneck. Currently, our method uses only 16 tokens per concept and performs well in 4-concept scenarios. Given the current context length of GPT4o(128K), Qwen(4k) and the ongoing trend of increasing context windows in vision-language model development, the mentioned context length limitation is not a significant concern for MC-LLaVA.
>
> **Q2**: Source of VLM location capability.
>
> Location information is integrated via synergistic three-fold sources:
> - **VLM Pretraining Priors**: This approach leverages the inherent spatial and OCR capabilities of the backbone network, linking visual markers to textual identifiers.
> - **Implicit Attention**: Visual features injected into personalized textual prompts allow the VLM to focus on the texture of specific concepts during the autoregressive generation process, implicitly aligning them with image regions to differentiate similar concepts.
> - **Explicit Visual Prompt**: We overlay numerical markers (following the SoM paradigm) in high-confidence regions and inject textual anchors into the prompts.
>
> **Q3**: Clarification of visual prompt.
>
> "Mark number 1" is a textual string inserted into the user prompt. It acts as a semantic anchor that corresponds to the visual numeric label overlaid on the input image by our personalized visual prompt. It explicitly guides the model's attention to the specific spatial region associated with the learned concept.

---

> ### Author Response · Authors · 2025-11-24
>
> Dear reviewer r43i, As the discussion deadline is approaching, we kindly invite you to share your thoughts and engage in discussion if possible. We truly appreciate the time and effort you've put into reviewing our submission. If there are any points you'd like us to clarify or discuss further, we are more than happy to assist. Thank you again for your valuable feedback and contributions. Best regards, Authors of submission 7804

---

### Official Review · Reviewer_FkwZ · 2025-11-01

**Soundness:** 3
**Presentation:** 3
**Contribution:** 3
**Rating:** 6
**Confidence:** 3

**Summary:**

This paper presents a vision-language model designed for multi-concept personalization. Unlike prior single-concept approaches, the proposed MC-LLaVA jointly learns multiple user-defined concepts through multi-concept instruction tuning, visual-based token initialization via k-means, and personalized visual prompts for grounding. Moreover, the authors also build a new multi-concept instruction dataset.

**Strengths:**

1. The paper addresses an important limitation of existing personalized VLMs by focusing on multi-concept personalization, a realistic setting that significantly broadens the applicability of personalized vision-language models.

2. The experimental results and ablation studies showing the effectiveness of the proposed method.

**Weaknesses:**

1.  The paper lacks analyses on key factors such as the number of concepts per image, backbone dependence, and the choice of (k) in k-means initialization, leaving the robustness and sensitivity of the method insufficiently explored.

2. It is unclear how the confidence thresholds τ and γ are determined or tuned. It also lacks ablation studies or criteria to justify their choice, making the robustness of the method with respect to these parameters uncertain.

3. While useful, the proposed MC-LLaVA seems a straightforward engineering combination of existing ideas (e.g., multi-concept joint training and visual prompt aggregation) for multi-concept setting.

4. Although several visual examples are shown, the paper does not include failure cases or discuss limitations such as concept confusion, hallucination, or overfitting to synthetic identifiers.

**Questions:**

The QA pairs and captions of the proposed dataset are primarily produced via GPT-4o, with limited human verification. This raises concerns about potential annotation noise, circular evaluation bias, and whether GPT-generated labels are truly 'ground truth'.

---

> ### Author Response · Authors · 2025-11-21
> **Rebuttal by Authors (Part 1)**
>
> We sincerely thank the reviewer for acknowledging the **important limitations we addressed** and the **significant performance improvements** of our method. We appreciate your insightful comments and would like to provide more elaborations on each point of your response.
>
> **W1**: Analysis of mentioned key factors.
> - **Backbone dependence**:  Following your suggestion, we extended MC-LLaVA to a stronger backbone: **LLaVA-OneVision (7B)**. As detailed in Tab. 3 of the global response, MC-LLaVA consistently enhances personalization capabilities in this advanced architecture. This demonstrates that our paradigm is **model-agnostic** and scales effectively with stronger base models.
> - **The number of concepts per image**: Our evaluation covers 2-4 concept scenarios (mentioned in Sec 4.1). In **Appendix E.2 (Table 8)**, we analyzed the computational overhead. Results show that scaling from 2 to 3 concepts increases inference latency only marginally (0.78s -> 0.84s), confirming MC-LLaVA owns high efficiency handling multi-concept scenarios.
> - **The choice of (k)**: As shown in Tab. 2 of the global response,  we provide performances of MC-LLaVA on different value of K. Ablation studies can successfully validate the effectiveness of the design of K-means. Performance improves with k and peaks around k =16. While adaptive k value selection is an interesting direction, empirical results show that a fixed k = 16 providing robust performance across diverse concepts. Following Occam's razor, we choose k = 16 among different datasets and concepts.
>
> **W2**: Analysis of confidence thresholds.
>
> The thresholds τ and γ were initially determined empirically. To address your concern, we performed a sensitivity analysis by varying these parameters significantly. As shown in the table below, varying τ and γ results in only minor fluctuations across tasks, demonstrating MC-LLaVA maintains high stability. This robustness stems from our aggregation strategy, where averaging location confidence maps smooths out noise, making the final prompt generation resilient to threshold shifts.
>
> | τ| γ| MC-LLaVA |MC-LLaVA|MC-LLaVA| Yo'LLaVA | MyVLM|
> |-|-|-|-|-|-|--|
> ||| Rec.| VQA Acc| VG| Rec.| Captioning|
> | 0.1  | 100/(256×256)| 0.819| 0.875| 0.681| 0.919| 0.966|
> | 0.32 | 100/(256×256)| **0.878**| **0.890** | **0.723** | **0.947** | **0.975** |
> | 0.5  | 100/(256×256)| 0.851| 0.881| 0.710| 0.931| 0.970|
> | 0.75 | 100/(256×256) | 0.833| 0.878| 0.696| 0.926| 0.967|
> | 0.32 | 500/(256×256)| 0.856| 0.883| 0.711| 0.927| 0.969|
> | 0.32 | 1000/(256×256) | 0.833| 0.878   | 0.693| 0.925| 0.968|
>
> **W3**: Technical novelty.
>
> We respectfully argue that multi-concept personalization is **non-trivial** and cannot be solved by simply combining existing ideas.
> - **Failure of simple combination**: As shown in Fig. 1 and Tab. 3, naively merging single-concept methods (e.g., Yo'LLaVA-M) leads to severe concept confusion and performance degradation.
> - **Specifically designs**: To solve this, we introduced specific innovations:  (1) **Inter-Concept Negative Sampling**: A novel joint training objective that pairs Concept A images with Concept B queries within the same batch . This forces the model to learn discrimination, a capability absent in standard joint training. (2) **Synergistic effect with textual and visual prompt**: We utilize masked visual tokens to initialize textual embeddings, implicitly injecting fine-grained visual semantics into the language embedding space. This synergizes with our Personalized Visual Prompt, which offers explicit spatial guidance. While the visual prompt directs the model's focus spatially, the token initialization conveys what intrinsic features match semantically. This combined implicit-explicit mechanism effectively bridges the visual-textual gap, significantly accelerating convergence and reducing reliance on negative samples (Fig. 5).
>
> **W4**: Discussion of failure cases.
> We agree that analyzing failures provides a wider view. We have added related discussions in the "**LIMITATION AND FUTURE WORK**" section in the Appendix.
> - **Generally visually similar scenarios**: Following your suggestion, we constructed a "Hard Case" dataset (5 scenarios with visually similar objects, shown in Fig. 15-16). As indicated in Table 1 of the global response, MC-LLaVA shows substantial superiority over RAP-MLLM. The recently introduced Personalization Toolkit [1] underperforms in this instance, primarily because it depends exclusively on general visual similarities without employing a discriminative training method.
> - **Discussion of limitation**: As shown in Fig. 17, we observe that distinguishing visually nearly identical objects (e.g., identical twins with no distinct accessories) remains challenging. The most advanced model (e.g., GPT-4o, Gemini 3) also can not separate one concept from another, leaving an open problem for future work.
>
> [1] Personalization Toolkit: Training Free Personalization of Large Vision Language Models, Arxiv 2025.

---

> ### Author Response · Authors · 2025-11-21
> **Rebuttal by Authors (Part 2)**
>
> **Q1**: Dataset quality assurance.
>
> The utilization of large models for data generation during the training process has become increasingly common in research fields [2, 3, 4].  We acknowledge the concern regarding synthetic data. However, our pipeline follows rigorous standards to ensure quality:
> - **Human-in-the-loop Verification**: Our dataset was not purely auto-generated. A team of expertised AI researchers collaboratively designed the collection and performed manual quality checks. We manually selected diverse questions and refined the answers.
> - **Mitigating Circular Evaluation Bias**: We used different, more complex prompts for testing generation versus training generation to prevent pattern memorization. Furthermore, our strong performance on **Visual Grounding (VG)**—which requires precise personalized understanding that LLMs cannot hallucinate—confirms that the model has truly learned the visual concepts, validating the dataset's utility.
>
> [2] Yo'LLaVA: Your Personalized Language and Vision Assistant, NIPS 2024.
>
> [3] Improving CLIP Training with Language Rewrites, NIPS 2023.
>
> [4] ShareGPT4V: Improving Large Multi-Modal Models with Better Captions, ECCV 2024.

---

> ### Author Response · Authors · 2025-11-24
>
> Dear reviewer FkwZ, As the discussion deadline is approaching, we kindly invite you to share your thoughts and engage in discussion if possible. We truly appreciate the time and effort you've put into reviewing our submission. If there are any points you'd like us to clarify or discuss further, we are more than happy to assist. Thank you again for your valuable feedback and contributions. Best regards, Authors of submission 7804

---

### Author Response · Authors · 2025-11-21
**Author Rebuttal by Authors**

## Global Response to All Reviewers

We sincerely thank all the reviewers for their insightful reviews and valuable comments, which are instructive for us to improve our paper further.

To cope with real-world complex multi-concept personalization, this paper proposes the first multi-concept personalization paradigm, MC-LLaVA. We designed a multi-concept instruction tuning framework, including novel dual personalized textual and visual prompt. Beyond our prompt design, joint training strategy introduces **a significant novelty**: using examples from other concepts as negative samples, which is effective and convenient for multi-concept scenarios. **Comprehensive experiments and module analysis are included. MC-LLaVA achieves state-of-the-art performance while keeping the minimum token length and without using hard negative samples, better modeling the real-world scenarios.**

The reviewers generally held positive opinions of our paper, in that the proposed method is **"effective", "sound", "addresses an important limitation of existing personalized VLMs"** and **"reduce the need for high-quality negative samples, which is essential"**. Our well constructed dataset is **"significant", "crucial" because "this field lacks a large dataset for training and evaluation"**. The writing of this paper is **"concise"** and the experiments results are **"significant"** and **"superior"**.

The reviewers also raised insightful and constructive concerns. We made every effort to address all the concerns by providing sufficient evidence and requested results. Here is the summary of the major revisions:

- **Technical novelty (Reviewer Fkwz, BhVC)**: We highlight our specific algorithmic designs: Inter-Concept Negative Sampling for discriminative learning and Visual Token Initialization for implicit semantic injection. We emphasize the synergistic effect of our Dual Framework, where explicit visual prompts and implicit textual features work jointly to bridge the visual-textual gap.
- **Dataset details (Reviewer Fkwz, BhVC)**: We elaborate on the Human-in-the-loop verification process used to ensure the high quality of our dataset and mitigate potential noise from synthetic generation. We also calculate the mismatch rate to confirm the alignment between personalized VQA performance and recognition.
- **Analysis of hyperparameters (Reviewer Fkwz, r43i)**: We newly add hyperparameters sensitivity analysis on backbone dependence, the choice of (k) in k-means initialization, the number of images per concept and the confidence thresholds τ and γ. These experiments collectively demonstrate the robustness of MC-LLaVA across diverse settings.
- **Discussion of concept confusion (Reviewer Fkwz, r43i)**: We conducted additional quantitative evaluations on a new curated "Hard Case" subset (visually similar concepts) to verify the disambiguation capability of our method. We also included a discussion on extreme cases (e.g., identical objects) and limitations in the Appendix to provide a comprehensive and objective view.
- **Discussion of related works (Reviewer BhVC)**: We clarify the fundamental methodological differences between our MC-LLaVA and mentioned Personalized Toolkit, emphasizing our advantage in visually similar concept scenarios. Maintaining the input prompt consistent, MC-LLaVA outperforms it across different personalized tasks.

The valuable suggestions from reviewers are very helpful for us to revise the paper to a better shape. We'd be very happy to answer any further questions.

Besides, shared Tables of the global response are listed as follows, and Figures are provided in original PDF.

Tab. 1 Evaluation on "Hard Case" subset.
| Methods|||||
|--|--|--|--|--|
|| Rec.| VG   | VQA Acc | QA Acc |
| **LLaVA+P (10^1)**| 0.533| 0.508| 0.785| 0.556  |
| **MC-LLaVA**| **0.866**| **0.717**| **0.882** | **0.700** |
| **RAP-MLLM**| 0.672| 0.669 | 0.774  | 0.688|
| **Personalized Toolkit**   | 0.665| 0.654 | 0.768  | 0.676  |

Tab. 2 Comparison with Personalized Toolkit.
| Methods| Yo'LLaVA |Yo'LLaVA| MyVLM|
|--|--|--|--|
|| Rec.| VQA Acc | Rec.|
| **MC-LLaVA (Origin)**| 0.947| 0.939   | 0.975|
| **MC-LLaVA (long prompt)**| **0.955**| **0.943** | **0.986**|
| **Personalized Toolkit**  | 0.949| 0.934   | 0.983|

Tab.3 Ablation on the value of K.
| K| MC-LLaVA |MC-LLaVA|MC-LLaVA| Yo'LLaVA | MyVLM|
|--|--|--|--|--|--|
|| Rec.| VQA Acc | QA Acc | Rec.| Captioning|
| 32  | 0.870| 0.886| 0.704| 0.941| 0.970|
| 16| **0.878**| **0.890** | **0.709**| **0.947**| **0.975** |
| 8| 0.866| 0.875| 0.705| 0.937| 0.959|

Tab.4 Evalution with different backbones.
| Different Backbones| MC-LLaVA |MC-LLaVA|MC-LLaVA| Yo'LLaVA | MyVLM|
|---|--|--|--|--|--|
|| Rec.| VQA Acc | QA Acc | Rec.| Captioning |
| LLaVA+P (10^1)| 0.573| 0.817| 0.553| 0.835| 0.207|
| LLaVA-OV+P (10^1)| 0.601| 0.912| 0.788| 0.938| 0.411|
| MC-LLaVA (LLaVA)| 0.878| 0.890| 0.709| 0.947| 0.975|
| MC-LLaVA (LLaVA-OV)| **0.904**| **0.933** | **0.841** | **0.955** | **0.980** |

---

### Meta-Review · Area_Chair_SpkB · 2026-01-07

**Summary:**

This paper proposes MC-LLaVA for multi-concept VLM personalization. After carefully reviewing the reviewers’ comments and the authors’ rebuttal, I concur with the overall assessment that the submission does not yet meet the acceptance bar of ICLR. Although the rebuttal adds ablations and clarifications, several core concerns regarding novelty, dataset quality, grounding evaluation, and generalization across more advanced and recent models remain unaddressed. Therefore, my final recommendation is to reject the paper in its current form.

**Reviewer Concerns:**

All reviewers expressed concerns about the novelty and contribution of MC-LLaVA. The rebuttal enhances the narrative, but does not solve these concerns strongly. The processes for manual modification and quality control of the dataset are still lacking, which affects the reliability of the dataset. Additionally, the version of the model (GPT-4o) used for automated data generation is relatively old, which may also impact the quality of the generated data. Reviewer BhVC strongly disagrees that coarse textual location descriptions are sufficient as a grounding signal for multi-object personalization. The rebuttal narrows scope to 2-4 concepts, but this weakens the broader “real-world multi-object” claim and does not fully resolve the concern. Additionally, the proposed personalized visual prompt during inference is to enhance recognition and grounding capabilities, thus fine-grained evaluation is necessary. The experiments are limited to the LLaVA series. Training on more advanced and recent VLMs is necessary to demonstrate the generalizability of the approach and its value in the current research landscape.

**Reviewer Scores:**

The initial reviewer scores are 6/4/4, with two reviewers being negative and one positive. The rebuttal does not sufficiently address the key issues, thus I think the scores might remain unchanged or even be lower.

---

### Decision · Program_Chairs · 2026-01-26

Reject